# A seawater desalination scheme for global hydrological models

Naota Hanasaki[1][2], Sayaka Yoshikawa[3], Kaoru Kakinuma[3][4][5], and Shinjiro Kanae[3]

[1]National Institute for Environmental Studies, 16-2 Onogawa, Tsukuba, 305-8506, Japan

[2]International Institute for Applied Systems Analysis, Schlossplatz 1, Laxenburg, A-2361, Austria

[3]Department of Civil and Environmental Engineering, Tokyo Institute of Technology, 2-12-1-M1-6 O-okayama, Meguro-ku, Tokyo 152-8552, Japan

10 [4]Center for Climate Systems Research, Earth Institute, Columbia University, 2880 Broadway, New York, NY 10025, USA

[5]NASA Goddard Institute for Space Studies, 2880 Broadway, New York, NY 10025, USA

*Correspondence to:* N. Hanasaki (hanasaki@nies.go.jp)

Abstract

**Abstract.** Seawater desalination is a practical technology for providing fresh water to coastal arid regions. Indeed, 20 the use of desalination is rapidly increasing due to growing water demand in these areas and decreases in production costs due to technological advances. In this study, we developed a model to estimate the areas where seawater desalination is likely to be used as a major water source and the likely volume of production. The model was designed to be incorporated into global hydrological models (GHMs) that explicitly include human water usage. The model requires spatially detailed information on climate, income levels, and industrial and municipal water use, which represent standard input/output data in GHMs. The model was applied to a specific historical year (2005) and showed fairly good reproduction of the present geographical distribution and national production of desalinated water in the world. The model was applied globally to two periods in the future (2011–2040 and 2041–2070) under three distinct socioeconomic conditions, i.e., SSP (Shared Socioeconomic Pathways) 1, SSP2, and SSP3. The results indicate that the usage of seawater desalination will have expanded considerably in 30 geographical extent, and that production will have increased by 1.4–2.1-fold in 2011–2040 compared to the present (from $2.8 \times 10^9$ m$^3$ yr$^{-1}$ in 2005 to $4.0$–$6.0 \times 10^9$ m$^3$ yr$^{-1}$), and 6.7–17.3-fold in 2041–2070 (from 18.7 to

48.6×$10^9$m$^3$ yr$^{-1}$). The estimated global costs for production for each period are 1.1–10.6×$10^9$ USD (0.002–0.019% of the total global GDP), 1.6–22.8×$10^9$ USD (0.001–0.020%), and 7.5–183.9×$10^9$ USD (0.002–0.100%), respectively. The large spreads in these projections are primarily attributable to variations within the socioeconomic scenarios.

## 1. Introduction

Water is vital to society. However, due to population increases, economic growth, and climate change, there is growing concern over the sustainability of water use around the world. There have been a number of studies regarding present and future worldwide water availability and use. Initially, the mean annual runoff (i.e., river discharge) was regarded as the primary renewable water resource (Vörösmarty et al., 2000; Oki et al. 2001, 2003). With increasing sophistication of global hydrological models (GHMs), the sources of water have been subdivided into various categories, such as rivers, reservoirs, lakes, groundwater, and others (Hanasaki et al., 2008a,b; Wada et al. 2011, 2014; Döll et al., 2014).

In this study, we focused on desalinated water derived from seawater, which has not been explicitly represented as a water source in GHMs, except in a few cases. Desalination is a technique for removing high concentrations of minerals and salts from saline and brackish water. Although it accounts for a marginal fraction at present (5.2 $km^3$ $yr^{-1}$ of the 3,926 $km^3$ $yr^{-1}$ total global water withdrawal in 2005 reported in the AQUASTAT database, http://www.fao.org/nr/aquastat/), desalination may play a greater role in the future for several reasons. First, water demand is rapidly increasing in arid and semi-arid areas where precipitation is limited, and on some islands and cities with limited catchment areas. Desalination is a practical measure for providing freshwater to such areas because it is completely independent of the natural hydrological cycle. Second, due to rapid technological advances, the costs of production have dropped rapidly, making it competitive with traditional water resources (Bremere et al., 2001; Ghaffour et al., 2013; Ziolkowska et al., 2015). These two factors have encouraged the rapid installation of desalination plants globally (see Figure S1). Indeed desalination is considered an adaptation measure for climate change for the freshwater sector (Jimenz Cisneros et al., 2014).

Desalinated water has been included in some previous global water resource assessments based on simulations. Oki et al. (2001) presented a global water resource assessment incorporating reported national desalinated water production for various countries. Oki et al. (2003) conducted future global water projections assuming that the present volume of desalination production would remain unchanged. As production of desalinated water is increasing rapidly, however, this assumption underestimates the future contribution of desalination. Wada et al. (2011) incorporated desalinated water as a water source in their GHM PCR-GLOBWB by spatially distributing the national volume of usage for areas within 40 km of the seashore. They assumed that the volume would increase in proportion to the population. The above examples represent reasonable simplifications for GHMs, but further refinement is necessary because desalination is increasing rapidly and is strongly connected to climate and socioeconomic conditions. Independently of GHMs, a few studies have projected future growth of desalination on a global scale. Bremere et al. (2001) projected the desalination capacity required for ten water-scarce countries in 2025 to meet their growing municipal water demands. They assumed that water use per person would remain

constant and that growth in municipal water demand would be proportional to population increase. Fichtner
GmbH (2011) reported a detailed future projection of desalinated water production in the Middle East and North
Africa (MENA). The projection shows marked increases in desalinated water production based on the assumption
of rapid and widely distributed uptake of Concentrating Solar Power Desalination technology throughout the
MENA region. Ouda (2014) projected future water supply and demand in Saudi Arabia under three scenarios,
taking into account desalinated water and treated wastewater. Kim et al. (2016) proposed an economic approach
to project future desalinated water production globally. The authors assumed that supply and demand of
desalination is determined by price mechanisms, and reported that total desalinated water would reach 250 km$^3$
yr$^{-1}$ at the end of the 21$^{st}$ century, Further studies are needed to examine the effects of socioeconomic variables,
such as growing water use per person, and to cover the whole globe because of the increase in areas where
desalination is utilized.

In this study, we developed a model to infer the geographical distribution and production of seawater desalination.
The model was designed to be incorporated into GHMs that are applicable for state-of-the art global long-term
projections. Because of the low specificity of GHMs (e.g., the typical spatial resolution is 0.5° × 0.5°), our model
is primarily focused on regions where seawater desalination is used as much as conventional water sources, such
as river water and groundwater. It is beyond the scope of this study to reproduce or project individual desalination
projects all over the world.

There were two key questions. First, what are the climatic and socioeconomic conditions where seawater
desalination is implemented on a large scale around the world? Identification of such conditions would facilitate
the development of a model to explain the present geographical distribution and production of seawater
desalination. Second, what would be the production of desalinated water in the future under various
socioeconomic scenarios? The model was developed and validated utilizing newly available global datasets.
Future projections were conducted based on a comprehensive global water assessment (Hanasaki et al., 2013a,b).
Both the distribution and production of seawater desalination were assessed for three socioeconomic scenarios to
meet increasing industrial and municipal water requirements.

This paper is structured as follows. Section 2 describes the data, model, and simulation settings. In Section 3, the
results of the model and simulation are presented along with a discussion regarding its performance in
reproducing the present distribution of desalination plants. The results of future projections are also shown.
Section 4 concludes the paper.

## 2. Methods

2.1 Desalination data

We first collected country-based information on desalination production for nine countries with the largest seawater desalination capacity in the world from the AQUASTAT database: United Arab Emirates (UAE), Saudi Arabia, Kuwait, Spain, Qatar, Libya, Bahrain, Israel, and Oman (hereafter Major Countries). We selected the year 2005 as the base year mainly due to the availability of socioeconomic data as discussed below. In addition, we collected municipality-based information for Saudi Arabia (KICP, 2011) and the UAE (RSB, 2013). In spite of our efforts, sub-national information for other countries was not found.

To obtain spatially detailed data, we used DesalData ([http://www.desaldata.com/](http://www.desaldata.com/)). As of 2014, DesalData included data for 17,335 individual desalination plants around the world. The total capacity of desalination reached 31.4 km$^3$ yr$^{-1}$ globally from all water sources. Individual records contain data such as plant status, source water type, user category, plant size, and geographical location (longitude and latitude). Note that plant size shows the capacity of plants, not the actual production. The ratio between the capacity and production varies by project.

Since the objective of this study was to develop a model that would be applicable to the global domain for long-term water scarcity projections, we focused on information regarding active major plants using seawater as source water. Currently, seawater is the primary source of desalination, amounting to 19.2 km$^3$ yr$^{-1}$, followed by brackish water, saline inland and river water, and others. First, we selected the major desalination plants to be included in our analyses. The detailed selection criteria and the rationale are shown in Table S1 and the Supplemental Text. Next, we selected 613 large plants using seawater as the water source. We examined the list of plants and excluded records if necessary. For example, DesalData contain some records indicating that plants in inland regions use seawater as the source, which is erroneous. A total of 54 plants were excluded from this study. These are listed in Table S2 with the reasons for their exclusion. Finally, we used a total of 559 large plants using seawater as the water source (hereafter Major Plants). Although this is less than 4% of total records (559 out of 17,335), as shown in Table S3, these plants account for 81% of the total capacity of active seawater desalination plants (12.7 km$^3$ out of 15.7 km$^3$). The number of plants and their total capacity are summarized in Table S3.

To determine the distances between desalination plants and major cities of selected countries, the geographical data of major cities were collected. We referred to the GeoNames database ([http://www.geonames.org](http://www.geonames.org)), which reports the geographical locations and populations of major cities around the world. We selected cities with populations larger than 100,000 (hereafter Major Cities).

## 2.2 Hydrometeorological data

At present, Major Plants are located mostly in arid regions. This makes sense because alternative water sources are limited in such conditions and costly seawater desalination becomes a practical option. To express aridity, the ratio of precipitation to potential evapotranspiration (hereafter Aridity Index, AI) was calculated globally. We used the WATCH Forcing Dataset (Weedon et al., 2011) as a global meteorological dataset for the historical period in question. This dataset covers the whole globe at a spatial resolution of $0.5° \times 0.5°$ and includes basic meteorological variables at 6-hour intervals, i.e., precipitation, air temperature, specific humidity, wind speed, air pressure, and shortwave and longwave downward radiation. Using the data from 1971 to 2000, we calculated the mean annual precipitation and potential evapotranspiration for each grid globally. The latter was calculated using the H08 global hydrological model (Hanasaki et al., 2008a,b).

For the future period, we used the climate projections obtained with a global climate model (GCM), namely, MIROC4-ESM-CHEM. The selection and combination of the GCM and climate scenarios and the methods of downscaling and bias correction were identical to those reported previously (Hanasaki et al. 2013a,b).

## 2.3 Socioeconomic data

For population and GDP, we referred to the Shared Socioeconomic Pathways (SSPs) Database provided by IIASA (https://secure.iiasa.ac.at/web-apps/ene/SspDb), which covers both the historical (1980–2005) and future periods (2005–2100) for 203 nations. The data for future periods came from model-based projections under five distinct socioeconomic views, i.e., SSP1 (Sustainability), SSP2 (Middle of the Road), SSP3 (Fragmentation), SSP4 (Inequality), and SSP5 (Conventional Development; O'Neill et al., 2014). GDP is shown in Purchasing Power Parity (PPP) in 2005 US Dollars (USD). This means that both monetary inflation over time and differences in price among nations or regions are removed from the GDP projections in this study.

For nationwide water use data, including desalinated water production, we relied primarily on the AQUASTAT database for the historical period in question. This database includes municipal and industrial water use data for 200 nations from 1960 to 2010 at 5-year intervals. Data are available for most countries in 2000 or 2005 but are largely missing for other years. For the future period, we used the projections of Hanasaki et al. (2013a). They estimated water withdrawal for irrigation, industrial, and municipal use over three periods (2011–2040, 2041–2070, and 2071–2100, centered on 2025, 2055, and 2085, respectively). Projections of industrial and domestic water withdrawal were obtained with a semi-empirical model using electricity production and population as primary driving forces and scenario-based parameters of improvements in water use efficiency. The parameters were set to be compatible with the storylines of the SSPs (e.g., slow improvements in SSP3, rapid improvements in SSP1). Consequently, as shown in Figures 12 and 14 of Hanasaki et al. (2013a), the projected

future water withdrawal was largest in SSP3 and smallest in SSP1. Note that these projections are demand-based without specifying water sources. The key socioeconomic factors are summarized in Table 1. In this study, we focused on three scenarios, i.e., SSP1, SSP2, and SSP3.

2.4 Analyses of collected data

In this subsection, the collected data are analyzed to model seawater desalination around the world. We quantitatively investigate desalination production and usage at the national and subnational levels. Because seawater desalination is mainly used in coastal arid regions, aridity conditions and distances from the seashore to plants and cities are also examined.

Table 2 summarizes desalinated water capacity and production in the Major Countries. The countries accounted for 85% of the seawater desalination plant capacity in 2005. UAE and Saudi Arabia produced the largest volumes of desalinated water, which accounted for more than half of the global total. The Major Countries share two characteristics. First, all are located in the Middle East or on the coast of the Mediterranean Sea and have arid or semi-arid climates. Second, their income is relatively high: per person GDP exceeds 14,000 USD PPP person$^{-1}$ yr$^{-1}$.

Next, we focused on water use in the Major Countries, as shown in Table 2. More than three times the volume of water was used by municipalities than by industry, except in Spain. Globally, 99% of Major Plants were installed for combined municipal and industrial use, with 90% for municipal water use. Seawater desalination was seldom used for agricultural water supply. The production to capacity ratio varied considerably among nations, ranging from 53% to 110%; the ratio sometimes exceeded 100%, probably due to inconsistencies in the data and the year of report.

Figure 1 shows the geographical distributions of Major Plants in the Mediterranean and Middle East regions where the majority of plants are concentrated. Note that the plants were aggregated into gridded data with a spatial resolution of 0.5° × 0.5° for consistency with the final simulation. As clearly shown in the figure, the Major Plants were mainly located on the seashore and near large cities.

Figure 2 shows the AI of the present climate. Among the Major Countries, countries located in the Arabian Peninsula, namely, UAE, Saudi Arabia, Kuwait, Qatar, Bahrain, and Oman, are located in a hyper-arid climate, where the AI is below 0.15. The countries on the coast of the Mediterranean Sea, namely Spain, Libya, and Israel, are located in more humid climates than the above-mentioned countries, with AIs typically between 0.15 and 0.5. Note that the desalination plants in Spain are only located on the southeastern coast and some islands where the AI is substantially lower than that in the remaining regions. The fraction of desalinated water production to total industrial and municipal water withdrawal is shown in Table 2. The fraction for the countries in the Arabian

Peninsula exceeds 50% except for Saudi Arabia, while that for the Mediterranean countries falls below 20%. This indicates that desalinated water is a major water source for industrial and municipal water in countries in the Arabian Peninsula, while it is supplemental in the countries on the Mediterranean Sea.

We plotted the relationship between the AI and plant capacity in each grid cell (Supplemental Figure S2), which shows a remarkable difference in the distribution of capacity by AI. We applied the segmented regression method and estimated the break point at AI = 0.082.

Figure 3 shows the distances of the major desalination plants from the seashore and the primary cities of the Major Countries. Desalination plants are typically located on the seashore supplying water to cities at close range. Therefore, we considered that the summation of the distance between the seashore and a plant, and that from the plant to the closest Major City was the distance that seawater was transferred. We calculated the distance using the Clark Lab's IDRISI Taiga Release 16.05 GIS platform software (Eastman, 2009). Note that due to technical limitations of the software, the individual plants were first aggregated into 5' × 5' grid cells, and then the distance was estimated at intervals of 1 km. We analyzed seven of the Major Countries including cities larger than 100,000 persons. The results indicated that 90% of the major desalination plants were located within 15 km from the seashore and 170 km from Major Cities. Taking the distance from the coastline and cities into account, the area using seawater desalination extended approximately 170–185 km (roughly corresponding to the edge (165 km) and diagonal (233 km) of 3 × 3 grid cells with a spatial resolution of 0.5° × 0.5°) from the seashore.

For further geographical breakdown, Table 3 shows the water supply of the Major Cities in UAE and Saudi Arabia. The two largest cities in UAE, Abu Dhabi and Dubai, depend heavily on desalinated water. Municipal and industrial water for Abu Dhabi is supplied entirely by desalinated water (EAAD, 2012) and their plant capacity reaches $1.18 \times 10^6$ m$^3$ day$^{-1}$. Information on detailed water supply in Dubai was not available, but a news article suggested that 98.8% of Dubai's water is supplied by desalination (Hackley, 2013). The total desalinated water production for the six cities in Saudi Arabia shown in Table 3 accounted for 80% of the national desalinated water supply. For these cities, desalinated water accounted for as much as 69% of the total supply. The sources of water varied between cities and regions. Jeddah, Makkah, Al Taif, and Al Madinah were largely dependent on desalinated water. Here, we note two exceptional cities. Although located 450 km from the seashore, Riyadh obtains about half of its water from seawater desalination transferred via pipeline. In contrast, although located on the seashore, Dammam depends on seawater desalination for less than 50% of its water supply.

2.5 Seawater desalination model

We developed a Seawater Desalination Model (SDM) to estimate the areas where desalinated water is used and the volume of desalinated water production. As the model is intended to be incorporated into the global water

resources model H08 (Hanasaki et al., 2008a,b), for consistency, its spatial resolution was set at 0.5° × 0.5°.

SDM consists of a set of climatological, geographical, and socio-economic conditions to estimate the spatial extent of where seawater desalination is likely to be used. SDM identifies the grid cells in which all of the conditions given below are met. We called these grid cells Area Utilizing Seawater Desalination (AUSD) cells. In the AUSD grid cells, fresh water is produced by seawater desalination technology and used to meet local water demands under several assumptions. By combining the gridded map of AUSD and requirements for water withdrawal (i.e., potential water demand), which is standard input/output data of global hydrological models, we can estimate the global production of seawater desalination. Note that AUSD indicate where desalinated water is used and, hence, they are not identical to the location of individual desalination plants. It is expected that the spatial extent of AUSD includes the locations of major seawater desalination plants, but desalination plants are not necessarily located at every AUSD grid cell because desalinated water can be transferred to surrounding grid cells through the pipeline network. The conditions and assumptions of SDM were determined based on analyses of the national and subnational statistics of desalinated water production and the geographical distribution of desalination plants, as described in Section 2.4.

In this study, we developed SDM and its variant (hereafter SDM2). SDM2 has an identical model structure to SDM, but it consists of different objectives and sets of conditions and assumptions. SDM estimates AUSD where industrial and municipal water are mostly dependent on seawater desalination. As seen in Section 2.4, countries such as UAE and Qatar are typical cases. SDM2 estimates AUSD where industrial and municipal water are at least partly dependent on seawater desalination. Countries on the Mediterranean Sea are typical cases.

SDM extracts AUSD or the grid cells meeting the following three conditions:

**Condition A)** Nations whose GDP exceeds 14,000 USD PPP person$^{-1}$ yr$^{-1}$.

**Condition B)** AI below 8%.

**Condition C)** Three consecutive grid cells (approximately 165 km along the equator) of seashore.

In general, the desalination plants and the regions using their production are located in arid coastal zones in relatively high-income countries. The globally uniform thresholds were determined by the analyses given in Section 2.4. Table 2 shows that GDP exceeded 14,000 USD PPP person$^{-1}$ yr$^{-1}$ in all of the Major Countries. Figure S2 shows that there is a breakpoint of desalination capacity at an AI of 8.2%, but it was rounded to 8%. Figure 3 shows that 90% of Major Plants are located 15 km from the seashore and 170 km from Major Cities, implying that seawater is transferred a distance of approximately three grid cells from the seashore. The thresholds are dependent on the climatological data used and spatial resolution of interest; hence, is expected to change if SDM is applied with different data at a different resolution. Several important factors relating to the design of the actual plants, such as the geographical suitability of seawater intake (Voutchkov, 2012), were excluded from this model

because of the model's indefinite resolution and the lack of available global data.

SDM estimates desalinated water production under the following assumptions:

**Assumption A)** Seawater desalination is used for municipal and industrial purposes, not for irrigation.

**Assumption B)** All municipal and industrial water withdrawal in AUSD is supplied by seawater desalination.

**Assumption C)** The production to capacity ratio, which shows the fraction of the seawater desalination production of a country relative to the total plant capacity is 30–80%.

Combining these assumptions, AUSD, and grid-based requirement of water withdrawal, the volume of desalinated water production can be estimated, which is directly transferable to global hydrological models. The rationale of Assumption A is that desalinated water is not considered to be affordable for irrigation (e.g., Bremere et al., 2001). Note the production cost is approximated as 0.86–3.21 USD m$^{-3}$ (Lamei et al., 2008). Also, this is supported by our findings from Table 2. Assumption B is linked with Condition B: under such arid climatic conditions, surface

water is hardly available unless major rivers are flowing (e.g., the River Nile in Egypt) and groundwater is not abundant over the long term because natural recharge must be virtually zero. This also implies a certain economic advantage of seawater desalination over alternative water sources, although we did not explicitly include the cost for each water source into our model. Assumption C is based on the findings of data analyses shown in Sect. 2.4. Note that the original range of the production-to-capacity ratio was 28–77%, shown in Table 3, but it was rounded to 30–80%.

SDM2 is a variant of SDM. SDM2 replaces Condition B of SDM with an AI below 50% and adds a new Condition D, which is that the withdrawal-to-water-resources ratio (WWR) is above 40%. WWR is the ratio of annual total water withdrawal of all sectors to the mean annual renewable water resources typically substituted by annual runoff. Both AI and threshold WWR were devised by Raskin et al. (1997), and are widely accepted and

280 used in quantitative macroscale water resource assessments (e.g., Vörösmarty et al., 2000; Oki et al., 2001). With these new conditions, SDM2 is able to identify the coastal regions with modest aridity and high water stress where actual major desalination plants are currently located.

Unlike SDM, it is difficult for SDM2 to maintain Assumption B. Outside hyper-arid climatic areas, in many cases, several water resource options are available, and seawater desalination only provides supplemental water. To estimate the desalination production, it is necessary to first identify how much of water withdrawal is assigned to seawater desalination. This could be partly achieved if SDM2 and GHMs such as H08 were fully coupled, but such modeling and simulations require considerable additional work and are beyond the scope of this paper. Therefore, SDM2 was only used to estimate AUSD, and not the volume of desalination.

 2.6 Simulations

Four simulations were performed using SDM and SDM2. The first was a historical simulation for 2005, which was used for model validation. The results were used to validate the locations and production volumes of desalinated water. The remaining simulations were simulations of SSP1, SSP2, and SSP3 for three periods (2011–2040, 2041–2070, and 2071–2100), which were consistent with the projections of Hanasaki et al. (2013a, b). The results were used to assess future desalinated water use.

## 3. Results and Discussion

### 3.1 Historical simulation

Historical simulations were performed to validate SDM and SDM2. First, we validated the geographical extent of
300 the simulated AUSD. Validation was conducted in two steps. First, we checked whether AUSD existed in the Major Countries shown in Table 2. Next, we checked whether the AUSD included the grid cells with desalination plants shown in Figure 1. The extent of AUSD is expected to be greater than the area of desalination plants, because desalinated water produced in a grid cell can be transferred to multiple cells through the pipeline system. Figure 4a shows the AUSD derived from SDM. Recall that the objective of SDM was to identify the regions dependent, to a significant extent, on seawater desalination. The AUSD included the location of Major Plants in the Middle East (Figure 1), spreading along the seashore in Kuwait, Saudi Arabia, UAE, Bahrain, Qatar, and Oman. Yemen was not included because it fell below the income threshold, which agreed well with Table 2 indicating that Yemen produces little desalinated water. Southern Oman was indicated as suitable for desalination, but this area had no desalination plants in 2005. This does not represent an overestimation of desalinated water
310 production as the region is sparsely populated, and therefore water withdrawal is quite limited compared to the northeastern part of the country. Few AUSD are found in the Mediterranean except for a part of Libya, which is consistent with the findings in Section 2.4 that the fraction of desalination production to water withdrawal was relatively low in this region.

The simulated AUSD of SDM2 are shown in Figure 4b. AUSD were extensive not only in the Arabian Peninsula but also in the Mediterranean region. This regions included the grid cells of the southeastern coast of Spain, western Mallorca, southern Sicily, and northwestern Libya, which agreed well with the actual distributions of major desalination plants (Figure 1). In general, the AUSD of SDM2 were concentrated near the large cities, as compared to that of SDM. Although SDM2 performed well, particularly in the Mediterranean coastal regions, the AUSD of SDM2 differed from the actual situation in some cases. First, AUSD expand in large parts of the coastal
320 regions of California in the USA (see Figure S5a). Desalination plants exist in the region, but they produced marginal volume in 2005. Although the grid cells for California were categorized as a water-stressed area under

the WWR index, in reality, water is effectively transferred long distances owing to California's long history of intensive water resource development. This implies that inter-grid water transfer should be taken into account when WWR is calculated. Another example is that, in reality, there are no plants at the southwestern edge of the Italian Peninsula, but they do exist in the central northern coast of Algeria; however, the AUSD of SDM2 show the opposite. The geographical distributions of desalination plants reflect various local circumstances and it is difficult to estimate AUSD grid by grid.

Because the stand-alone SDM2 model is unable to estimate the volume of production, the results from SDM are discussed hereafter. Table 2 includes the global and national volumes of desalinated water production for 2005 estimated by SDM. First, global total production and capacity of seawater desalination were estimated at $2.8 \times 10^9 \text{m}^3 \text{ yr}^{-1}$ and $3.5–9.4 \times 10^9 \text{m}^3 \text{ yr}^{-1}$, respectively. Note that the wide range in capacity reflects the production-to-capacity ratio included in Assumption C. The reported total global plant capacity in DesalData ($5.5 \times 10^9 \text{m}^3 \text{ yr}^{-1}$) was within the range of the latter. Next, the simulation results for national production agreed fairly well with AQUASTAT (Figure S2 shows a scatter plot of the data in Table 2). For Saudi Arabia, Kuwait, Qatar, and Oman, the simulated and reported desalinated water production agreed well with AQUASAT. The simulation for UAE underestimated the data from AQUASTAT. This was largely attributable to an inconsistency within AQUASTAT, in which desalination production exceeds the total of municipal and industrial water withdrawal. The severe underestimation in Spain and Israel was mainly due to lack of AUSD estimation; there were no AUSD in Spain, and only one grid cell in Israel. Note that we could not perform the calculations for Bahrain because the area of land could not be resolved in our model.

At this stage, SDM has been validated, mainly, based on the spatial extent of areas and total volume of desalinated water production in 2005. Although a systematic validation of the temporal dynamics of desalinated water was hampered mainly by the lack of access to long-term data on socio-economic conditions and water use, a simple validation was performed and is shown in Figure 5. Conditions A−C and Assumptions A−C in SDM indicate that AUSD vary over time, due to changes in municipal and industrial water withdrawal, because the other factors are unchanged during the time frame of a few decades. Figure 5 shows the relationship between national desalination capacity and municipal and industrial water withdrawal in AUSD for different periods for 1980–2005 (the methods summary is given in Supplemental Material). In general, the plots are located near the diagonal line. This supports Assumption B in SDM that the growth in municipal and industrial water in AUSD is sustained by seawater desalination. Exceptions are the cases of Spain and UAE. As we have mentioned several times above, AUSD in Spain are not well reproduced in this study. Although desalination capacity substantially increased in UAE during the period, AQUASTAT reported little increase in municipal and industrial water withdrawal. Taking into account the marked economic growth in UAE during the period, we inferred that water withdrawal would

have grown more than reported in AQUASTAT.

In summary, SDM was validated from three perspectives: geographical extent of AUSD (Figure 4a), total desalinated water production in 2005 (Table 2 and Figure S2), and temporal change in production (Figure 5). The simulated regions that rely largely on seawater desalination were well reproduced, as compared to earlier studies referred to in the Introduction Section. The regions are typically hyper-arid coastal regions where it is difficult to find alternative water sources. SDM also reproduced the reported volume of desalination production fairly well in the Major Countries. Because such countries dominate global water desalination, SDM also captured the total global volume reasonably well. Although SDM and SDM2 advance our resources for modeling global desalination, the results conveyed uncertainties, which can mainly be attributed to the limited availability of data and the simplicity of the model structure, similar to the future simulations described below. SDM2 was validated only for the geographical extent of AUSD. The performance was promising because it reproduced reasonably well the present distribution of Major Plants.

3.2 Future simulation

For future simulation, we mainly report the results of SDM because it produced outputs of both AUSD and the volume of desalinated water production. Figure 6 shows the estimated spread of AUSD at present and in 2055 under SSP1, SSP2, and SSP3 scenarios together with the actual distribution of extra-large desalination plants (capacity$\geq$ 50,000 m$^3$ d$^{-1}$) that meet the criteria shown in Table S1 in 2005 and 2014. For all cases in 2055, AUSD have expanded considerably compared to the present. The change in AUSD can be largely explained by growth in income , with the change in AI due to climate change playing a marginal role. In the case of SSP1, AUSD showed marked expansion of grid cells including Baja California in Mexico, northern Chile, the coastline of Northern Africa, and Southwest Africa. Note that the coastlines on the Caspian Sea in Central Asia became AUSD, as the inland lake is categorized as a sea. The expansions of SSP2 and SSP3 were somewhat limited compared to SSP1 because income levels in these scenarios were lower than in SSP1 (see Supplemental Figure S3). For example, the AUSD of SSP1 exclusively include Mauritania, Sudan, and Pakistan. The AUSD of SSP2 show no AUSD in Australia, which is attributed to wet climate under this scenario. As all SSP scenarios project considerable income growth with national income exceeding 14,000 USD PPP person$^{-1}$ yr$^{-1}$ for most countries in the world (Table 1 and Figure S3), AUSD are expected to expand into most of the arid areas in the world by 2055 (compare Figure 2 and Figure 6).

It is difficult to assess the validity of future projections, but comparing AUSD projections with the present distribution of desalination plants (shown in Figure 6e, f) can provide some insights. Interestingly, although

production in 2005 was not significant, several desalination plants were actually operational in Southern Africa and northern Chile, which is consistent with our AUSD expansion. In contrast, although at least one extra-large plant was located in Algeria, Angola, northern China, southeastern India, Japan, Korea, Singapore, Thailand, Florida in the USA, and Venezuela in 2014, AUSD did not cover these regions in any scenarios. This was primarily attributable to Condition B in SDM, which excluded non-hyper-arid regions. (As mentioned above, AUSD projections using SDM2 are shown in Supplemental Figure S5). AUSD of SDM2 expanded into Algeria, northern China, and southeastern India, which was consistent with the distribution in 2014. AUSD did not expand into other East and Southeast Asian countries because their AI was too high.

Using SDM, the projected levels of desalinated water production in 2025 and 2055 are 1.4–2.1 and 6.7–17.3 times greater than present, respectively (Table 4). The earlier estimates show a considerable range, but our results are within the range of spread for the reported period of 2015–2025. Only one earlier report was found with projections for the year 2050. Fichtner GmbH (2011) reported that total desalination production would be as high as 96,800 $\times 10^6$ m$^3$ yr$^{-1}$, which is more than double our estimate under SSP3. Their projection is based on the assumption that Concentrating Solar Power (CSP) Desalination, which is an emerging technology for producing desalinated water without fuel consumption, would extend widely throughout MENA. Although the technical problems are substantial for large-scale installation of CSP desalination plants, the development of new technology may lead to further increases in desalination production. Among the three SSPs, the volume of desalination is greatest in SSP3. This is primarily due to the higher water demand in SSP3, which is accompanied by higher population and lower efficiency (Hanasaki et al., 2013a). The considerable growth in desalination in SSP3 may be perceived as contradictory to its narrative scenario: SSP3 depicts a world of low income and inefficiency; i.e., one in which it will be difficult to adapt to and mitigate climate change. It is also true, however, that the country-based GDPPC grows steadily and considerably in all SSP scenarios. The world in SSP3 is poorer and more fragmented than in other SSPs, but that does not mean poorer than the present day.

Further regional breakdown is illustrated in Table 5. We followed the regional classification of the SSP scenarios, which subdivides the world into 11 regions (see Table S4 for the list of countries). In 2005, 85% of the production was concentrated in MENA (including all Major Countries except Spain). The second largest region of production was North America, which requires caution in its interpretation. A total of 409 $\times 10^6$ m$^3$ yr$^{-1}$ of seawater desalination in this region was attributed to the AUSD of seven grid cells in Southern California, USA. In particular, two grid cells including the suburbs of Palm Springs and El Centro were responsible for most of the production. In reality, there was no seawater desalination in these regions as of 2005, and SDM failed to reproduce the real situation for these cases. This does not necessarily mean that SDM is completely unreliable; in fact, several very large desalination plants are under planning in surrounding areas (e.g., the Carlsbad

Desalination Plant in San Diego started operating in 2015). MENA was projected to continue producing the largest volume of seawater desalination in 2025. Several new regions appear in the table, notably Latin America
and the Caribbean. The emergence of these countries is mainly due to the per person GDP of Mexico, Chile, and Peru exceeding the threshold of 14,000 USD PPP person$^{-1}$ yr$^{-1}$. In 2055, in SSP1, per person GDP exceeds the threshold in two densely-populated countries, Egypt and Pakistan, which caused a marked increase in seawater desalination in MENA and Southern Asia (ibid). Again, interpretation of these results requires caution. The AUSD in Egypt and Pakistan included grid cells through which the Nile and Indus Rivers, respectively, flow. Although these grid cells are located in extremely dry conditions, it is likely that water is supplied from these rivers to the cells. Assumption B would be not valid for these grid cells; hence, the AUSD may be overestimated. To improve the reliability of the projections, the availability of alternative water resources should be evaluated at each grid cell. Such simulations should be implemented when the SDM is fully integrated into global hydrological models with detailed water-use sub-models, such as the H08 model (Hanasaki et al., 2008a,b).

The economic cost of seawater desalination was estimated for each case (Table 4 for worldwide, Tables S5 and S6 for regions). Adopting the method of Lamei et al. (2008), we estimated the unit production costs at 0.40–3.78 USD m$^{-3}$ based on the data available for 186 plants globally (see Supplemental Material for details). As shown in Table 4, the volume of seawater desalination was estimated at $2.8 \times 10^9$ m$^3$ yr$^{-1}$, with a cost of $1.1–10.6 \times 10^6$ USD, equivalent to 0.002%–0.019% of total global GDP. This number is much higher in MENA, with values of 0.032%–0.306%. Among the scenarios, SSP3 in 2055 showed the highest numbers at 0.011%–0.100% globally and 0.121%–1.147% for MENA. This was mainly attributable to large water use and low GDP growth in SSP3.

The future projection is sensitive to Condition A (threshold of 14,000 USD PPP person$^{-1}$ yr$^{-1}$), Condition B (aridity index of 0.08), and Condition C (three grid cells of seashore). Here, we conducted a sensitivity test on these thresholds. We examined thresholds of 0, 7,000, and 24,000 USD PPP person$^{-1}$ yr$^{-1}$ for Condition A. A
threshold of 7,000 USD PPP person$^{-1}$ yr$^{-1}$ implies further reduction in cost due to technological change, while 0 implies that seawater desalination economically outperforms traditional water resource development in arid regions. The threshold of 24,000 USD PPP person$^{-1}$ yr$^{-1}$ was taken from the per capita GDP of Israel, one of the non-oil-producing nations in Table 2. Present Condition A was taken from the per capita GDP of Libya, which is an oil-producing nation. We considered that the threshold for oil-producing nations may be lower than for other countries due to massive consumption of fossil fuels for desalination. Similarly, we tested thresholds of 0.15 and 0.2 for Condition B, which expanded the potential area for AUSD (see Figure 2), and two and four grid cells of seashore for Condition C.

The results of the sensitivity tests are shown in Figure 7. First, a higher threshold of per capita GDP decreases the total desalination production due to reduction in AUSD. For the base year (2005), there is a gap between 0 and

450 7,000 USD PPP person$^{-1}$ yr$^{-1}$. In contrast, there is no change in SSP1 between 0 and 14,000 USD PPP person$^{-1}$ yr$^{-1}$, as per capita GDP exceeds 14,000 for most countries (see Figure S4). The large drop between 14,000 and 24,000 USD for SSP3 indicates that in this scenario a large number of countries cannot exceed a per capita GDP of 24,000 by 2055 (i.e., the growth in per capita GDP in SSP3 is considerably slower than in the other scenarios). The discrepancies among SSPs are attributable to differences in water use. Hanasaki et al. (2013a) projected the lowest water use for SSP1 due to rapid decreases in municipal and industrial water use intensity, with the opposite being true for SSP3. The sensitivity test of AI showed a linear response (only the results of SSP1 are shown in Figure 7) due to gradual changes in AI (see also Figure 2) and grid cells of seashore. Desalination production expands by increases in AI and the grid cells of seashore. The expansion was not proportional to the change in AI and the number of grid cells reflecting the complex geographical distribution of AI and shape of coastline.

3.3 Implications in the context of global water resource assessment

In 2005, the global total for industrial and domestic water withdrawal was reported to be 1,170 km$^3$ yr$^{-1}$(AQUASTAT). In this study, desalination production was estimated at 2.8 km$^3$ yr$^{-1}$ for these two purposes, accounting for 0.24% of the total. As shown in Table 1, water withdrawal will change considerably in the future. In the case of SSP1, which depicts a sustainable world, the projected water withdrawal is 1,142 km$^3$ yr$^{-1}$ in 2055, showing almost no change from the present. However, in SSP3, which depicts a fragmented world, withdrawal will more than double and reach 2,831 km$^3$ yr$^{-1}$. The estimated seawater desalination production levels are 18.7 and 48.6 km$^3$ yr$^{-1}$ for these scenarios, accounting for 1.6% and 1.7% of the global requirement, respectively. Although this portion appears small, desalinated water plays a critical role in AUSD. The population in the AUSD
is estimated at 12.8 million in 2005, but by 2055, is expected to increase to 138 million in SSP1, and 170 million in SSP3. The substantial increase in number was mainly attributable to AUSD extending into densely populated northern Africa and part of South Asia. As the AUSD are located in extremely dry regions, vulnerable to water scarcity, it is crucial to secure municipal and industrial water for water security and human well being in AUSD regions. Moreover, the estimated geographical extent of present and future AUSD largely overlap with regions where water resources are projected to decrease due to climate change, typically in the Mediterranean (Jiménez Cisneros et al., 2014).

3.4 Key uncertainties and limitations

The SDM model estimates AUSD and the volume of desalinated seawater based on three conditions and three
assumptions. The results demonstrate that these conditions and assumptions allow us to successfully simulate seawater desalination in accordance with the purpose of this study. Here, each condition and assumption is

revisited and key uncertainties are discussed.

Conditions A and B, which set thresholds on national average income and aridity, explain the present distribution of major desalination plants and regions that are dependent on seawater desalination. It should be noted that countries meeting these conditions include major oil-producing countries. Seawater desalination has high energy consumption costs, even with the application of the latest technology. Hence, the price of energy is an important factor in the installation of desalination plants. Although we confirmed that non-oil-producing countries such as Spain and Israel had considerable desalination capacity, attention should be paid to the projected newly emerging AUSD in non-oil-producing countries under the SSP scenarios, such as Chile, Namibia, etc. The income threshold

may differ for these nations compared to major oil-producing countries because of differences in energy prices. These regional biases cannot be completely eliminated from our model simulations and validation because the majority of desalination plants have, so far, been constructed in the Middle East. It should also be noted that we fixed the thresholds for income and aridity according to Conditions A and B throughout the study period. This corresponds to the assumption that technology and costs of production are essentially fixed at present. Advances in technology would likely further lower the production costs, which would alter the threshold for plant installation (e.g., Ghaffour et al., 2013; Ziolkowska et al., 2015).

Condition C, which sets the maximum distance from the seashore for AUSD, plays a crucial role in the simulated volume of desalinated water. Desalinated water can be used at large distances from the seashore if transportation costs are affordable. Assumption B, or 100% dependence on desalination in AUSD, is probably the largest

uncertainty in the SDM. Although surface water is unreliable under super-arid conditions in AUSD, the dependence is influenced by the availability of other water sources, typically rivers, groundwater, and sometimes long-distance water transfer (Lamei et al. 2008). Recycled water is another emerging source of freshwater in water-scarce regions. For example, Singapore has been strongly promoting recycled water usage, which is economically more efficient than seawater desalination due to Singapore's stringent waste water quality control and appropriate infrastructure (Tortajada, 2006). In contrast, major desalination plants are occasionally implemented in relatively wet regions, e.g., on islands that are characterized by relatively small catchment areas, limited storage capacities, and large temporal variations in water demand. Such regional details are beyond the capability of present global hydrological models, but may substantially affect the results in some places.

In this study, we used the future water use projection of Hanasaki et al. (2013a) based on empirical relationships

between water use and population and electricity production excluding regional details, and therefore projections for individual nations include substantial uncertainty. The spatial distribution of water use is another source of uncertainty. Historical and future water use are obtained first at the national scale, and then converted into grid cells under various assumptions. This study adopted the assumption that the spatial distribution of municipal and

industrial water use is proportional to that of population. We note that supply and demand are interconnected: seawater desalination relieves the availability of water constrained by hydrology. Increases in water availability may enhance water consumption. This may alter the future situation substantially from the historical past on which the water demand projections of Hanasaki et al. (2013a) were based. Water-use projections substantially differ among models and a systematic model inter-comparison is under way (Wada et al. 2016).

Finally, we summarize the limitations due to the preconditions of the SDM model. First, SDM was designed to be incorporated into GHMs, typically simulated at a spatial resolution of $0.5° \times 0.5°$ (approximately $55 \times 55$ km at the equator). It is impossible to resolve individual plants with this spatial scale; hence, this is outside the scope of this model. Second, although the cost of desalination and its difference in cost to alternative sources play a crucial role in desalination projects in reality, this price mechanism is not included in the present formulation of SDM. Third, partly as a consequence of the previous factor, SDM is unable to analyze the effects of technological advances and subsequent cost reductions on the introduction of new desalination plants to regions with wetter climates and lower incomes. The results of the sensitivity test of changing the threshold of Condition A (minimum per capita GDP; Figure 7) and Condition B (maximum AI) provided us with several useful insights, but further investigations are necessary to assess the combined effect. Fourth, the availability of alternative water resources cannot be estimated by the stand-alone SDM model. Such analyses would be feasible

if we combined SDM2 with a global hydrological model that explicitly simulates renewable/fossil groundwater. Fifth, the thresholds in SDM were fixed throughout the simulation period. The above-mentioned factors, namely, the relative cost of desalination relative to alternative water sources, technological advances in desalination, and availability of alternative water sources, might change the thresholds in the future. Sixth, the conditions and assumptions used for future simulations were independent of the narrative scenarios of the SSPs. Reflecting the conditions and assumptions in the narrative scenarios would enhance the consistency between our study and the SSPs (e.g., SSP1 depicts a world of sustainability; hence, conventional energy-intensive desalination plants would not be favored). Seventh, in general, the performance of SDM is highly dependent on the availability of data. In this study, we mainly used datasets with global coverage (DesalData and AQUASTAT), and reports published by academic and governmental institutions

written in English. Reports published by the private sector, and written in local languages were not collected due to limitations in capability and resources. International collaboration would be needed to enable systematic collection of further data.

## 4. Conclusions

We have developed two models to estimate the location and volume of desalinated water production. First, we identified climatic and socioeconomic conditions that are common to areas where seawater desalination is undertaken. Three typical conditions were found, i.e., relatively high income, aridity, and proximity to the seashore. We obtained common global parameters for each, and demonstrated that the present AUSD can be fairly well reproduced by the proposed model. Then, we assumed that municipal and industrial water in AUSD are fully supplied by seawater desalination, and estimated the production and plant capacity of seawater desalination. We achieved fairly strong agreement with independent data. Second, using the SDM and SSP socioeconomic scenarios, the future production of desalination water was projected globally. The results indicated that AUSD are expected to expand considerably in the 21$^{st}$ century. Income growth plays a primary role in the expansion of desalination plants.

Desalination is a practical engineering measure for meeting the growing water demand in arid regions and for adapting to climate change (Jiménez Cisneros et al., 2014). This study proposes one of the first models to express desalination in global water resource models supported by the available literature and technologies. Although further improvements are needed, the model provides a good starting point for dynamically incorporating information regarding desalination into global water resource assessment.

## Acknowledgements

This work was mainly supported by CREST, Japan Science and Technology Agency. N. Hanasaki acknowledges the support of JSPS KAKENHI Grant Number 25820230 and the Environment Research and Technology Development Fund (S-14) of the Ministry of the Environment, Japan. S. Yoshikawa, K. Kakinuma, and S. Kanae acknowledge the support of JSPS KAKENHI Grant Number 15H04047 and 16H06291. The authors are grateful to three anonymous reviewers, Yoshie Maeda, and Yaling Liu for helpful suggestions. The present work was partially developed within the framework of the Water Futures and Solutions initative at IIASA and the Panta Rhei Research Initiative of the International Association of Hydrological Sciences (IAHS) by the Water Scarcity Assessment: Methodology and Application working group. Map colors are based on www.ColorBrewer.org, by Cynthia A. Brewer of Pennsylvania State University.

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

Table 1. Global future scenarios in 2055 (extracted from Hanasaki et al., 2013a, b)

| Item | 2005 | 2025 | | | 2055 | | |
| --- | --- | --- | --- | --- | --- | --- | --- |
| | | SSP1 | SSP2 | SSP3 | SSP1 | SSP2 | SSP3 |
| Radiative forcing | - | RCP2.6 | RCP4.5 | RCP6.0 | RCP2.6 | RCP4.5 | RCP6.0 |
| Global mean temperature rise [K] | 0 | 1.7 | 1.5 | 1.4 | 2.4 | 2.9 | 2.9 |
| Population [$10^9$person] | 6.5 | 7.4 | 7.8 | 7.9 | 8.5 | 9.3 | 10.3 |
| GDP [$10^{12}$USD] | 56 | 88 | 84 | 79 | 319 | 255 | 186 |
| Municipal water withdrawal [$km^3$/yr] | 446 | 544 | 599 | 632 | 622 | 823 | 936 |
| Industrial water withdrawal [$km^3$/yr] | 724 | 853 | 1169 | 1436 | 520 | 1437 | 1895 |

Table 2. Major countries producing desalinated seawater in 2005.

| | Country | GDP\|PPP per person (SSP DB) [2005 USD person⁻¹ yr⁻¹] | Total water withdrawal (AQUASTAT) [10⁶m³ yr⁻¹] | | Desalinated water production [1] (AQUASTAT) [10⁶m³ yr⁻¹] | | Desalination water plant capacity (DesalData) [10⁶m³ yr⁻¹] | | | | Production to withdrawal ratio [%] | Production to capacity ratio [%] | Desalinated water production (simulated) [10⁶m³ yr⁻¹] | | | Desalination water plant capacity (simulated) [10⁶m³ yr⁻¹] |
|---|---|---|---|---|---|---|---|---|---|---|---|---|---|---|---|---|
| | | | Mun(A). | Ind(B) | Total (C) | | Mun. | Ind. | Others | Total (D) | C/A+B | C/D | Mun. | Ind. | Total | Total |
| 1 | UAE | 66,855 | 617 | 69 | 950 | (2005) | 1,760 | 46 | 0 | 1,806 | 139 | 53 | 617 | 69 | 686 | 858–2,287 |
| 2 | Saudi Arabia | 20,406 | 2,130 | 710 | 1,033 | (2006) | 1340 | 129 | 0 | 1,469 | 36 | 70 | 640 | 213 | 853 | 1,066-2,843 |
| 3 | Kuwait | 48,783 | 448 | 23 | 420 | (2002) | 448 | 0 | 0 | 448 | 89 | 94 | 448 | 23 | 472 | 590–1,573 |
| 4 | Spain | 27,379 | 5,790 | 6,698 | 100 | (2005) | 285 | 0 | 35[2] | 321 | 0.8 | 32 | 0 | 0 | 0 | 0 |
| 5 | Qatar | 69,512 | 174 | 8 | 180 | (2005) | 171 | 0 | 0 | 171 | 99 | 105[3] | 174 | 8 | 182 | 228–607 |
| 6 | Libya | 14,015 | 610 | 132 | NA[4] | NA[4] | 104 | 37 | 0 | 142 | NA | NA | 32 | 7 | 39 | 49-130 |
| 7 | Bahrain | 28,068 | 178 | 20 | 102 | (2003) | 113 | 24 | 0 | 137 | 52 | 74 | NA[5] | NA[45] | NA[3] | NA |
| 8 | Israel | 24,543 | 712 | 113 | 140 | (2007) | 135 | 0 | 0 | 135 | 17 | 104 | 10 | 2 | 12 | 15–40 |
| 9 | Oman | 21,047 | 134 | 19 | 109 | (2006) | 95 | 4 | 0 | 99 | 71 | 110 | 128 | 18 | 146 | 183–487 |
| | Others | | | | | | 459 | 308 | 0 | 802 | | | 98 | 319 | 417 | 519–1,383 |
| | Total | | | | 5,162 | | 4,911 | 584 | 35 | 5,530 | | | 2,147 | 659 | 2,806 | 3,507-9,353 |

1) Source water not specified.

2) Used for irrigation.

3) MDPS (2016) reports that desalination capacity and production in Qatar in 2012 are $1.42 \times 10^6 m^3 day^{-1}$ and $1.20 \times 10^6 m^3 day^{-1}$ respectively; consequently, the production-to-capacity ratio is 85% for this reference.

4) Data only available for 1998 and earlier.

5) Bahrain could not be resolved with the current spatial resolution of $0.5° \times 0.5°$.

Table 3. Major cities that use desalinated water

| City | Water supply [m³ day⁻¹] | | | | Desalination water plant capacity (DesalData) (B) [m³ day⁻¹] | Production to capacity ratio (A/B) [%] | Source |
|---|---|---|---|---|---|---|---|
| | Total | Groundwater | Desalinated water (A) | (year) | | | |
| Riyadh (Saudi Arabia) | 155,7000 | 745,000 | 812,000 | (2009) | 1,795,099 | 45 | KICP, 2011 |
| Jeddah (Saudi Arabia) | 633,000 | 3,000 | 630,000 | (2009) | | | KICP, 2011 |
| Makkah (Saudi Arabia) | 280,000 | 0 | 280,000 | (2009) | | | KICP, 2011 |
| Al Taif (Saudi Arabia) | 137,000 | 23,000 | 114,000 | (2009) | | | KICP, 2011 |
| Total of above three cities | 1,050,000 | 26,000 | 1,024,000 | (2009) | 2,557,833 | 40 | |
| Al Madinah (Saudi Arabia) | 327,000 | 55,000 | 272,000 | (2009) | 485,496 | 56 | KICP, 2011 |
| Dammam (Saudi Arabia) | 345,000 | 190,000 | 155,000 | (2009) | 547,000 | 28 | KICP, 2011 |
| Total of above six cities | 3,279,000 | 1,016,000 | 2,263,000 | (2009) | | | |
| Abu Dhabi (UAE) | | | 1,180,000 | (2013) | 1,520,000 | 77 | RSB, 2013 |

Table 4. Projected future production of desalinated water

| | Period | Domain | Desalination production from seawater $[\times 10^6\,m^3\,yr^{-1}]$ | Installed capacity for seawater $[\times 10^6\,m^3\,yr^{-1}]$ | Installed capacity for all sources $[\times 10^6\,m^3\,yr^{-1}]$ | Production cost $[\times 10^6\,USD]$ | Production cost to GDP [%] |
|---|---|---|---|---|---|---|---|
| **Historical** | | | | | | | |
| This study | 2005 | Globe | 2,806 | 3,507-9,353[1] | -- | 1,122-10,606 | 0.002-0.019 |
| AQUASTAT | 2003 – 2007 | Globe | 5,162[2] | -- | -- | -- | -- |
| DesalData | 2005 | Globe | -- | 6,900 | 12,100 | -- | -- |
| Bremere et al. (2001) | 2000 | 10 countries[3] | -- | 1,680[4] | 2,660 | -- | -- |
| Lattemann et al. (2010) | 2006 | Globe | -- | 10,200 | 16,100 | -- | -- |
| Fichtner GmbH (2011) | 2000 | MENA[5] | 4,561 | -- | -- | -- | -- |
| **Future (near term)** | | | | | | | |
| This study (SSP1) | 2025 | Globe | 4,050 | 5,063-13,500 | -- | 1,620-15,310 | 0.001-0.012 |
| This study (SSP2) | 2025 | Globe | 5,344 | 6,680-17,814 | -- | 2,138-20,201 | 0.002-0.017 |
| This study (SSP3) | 2025 | Globe | 6,030 | 7,537-20,100 | -- | 2,412-22,793 | 0.002-0.020 |
| Bremere et al. (2001) | 2025 | 10 countries[3] | -- | 5,080[4] | 8,070 | -- | -- |
| Lattemann et al. (2010) | 2015 | Globe | -- | 22,500[4] | 35,800 | -- | -- |
| Fichtner GmbH (2011) | 2020 | | 35,780 | -- | -- | -- | -- |
| **Future (long term)** | | | | | | | |
| This study (SSP1) | 2055 | Globe | 18,702 | 23,378-62,342 | -- | 7,481-70,695 | 0.002-0.022 |
| This study (SSP2) | 2055 | Globe | 40,306 | 50,383-134,354 | -- | 16,123-152,358 | 0.006-0.060 |
| This study (SSP3) | 2055 | Globe | 48,638 | 60,797-162,126 | -- | 19,455-183,851 | 0.011-0.100 |
| Fichtner GmbH (2011) | 2050 | MENA[5] | 96,797 | -- | -- | -- | -- |

[1] Assuming a rate of operation of 30%–80%.

[2] From all water sources.

[3] Malta, Barbados, Singapore, Jordan, Yemen, Bahrain, Kuwait, Qatar, Saudi Arabia, Libya.

[4] Values for seawater were not reported. We assumed 63% of the source was seawater, taken from the estimate for 2006 by Lattemann et al. (2010).

[5] Algeria, Bahrain, Egypt, Gaza, Iran, Iraq, Israel, Jordan, Kwait, Lebanon, Libya, Morocco, Oman, Qatar, Saudi Arabia, Syria, Tunisia, United Arab Emirates, Yemen.

Table 5. Production of desalinated water by region [$10^6$ m$^3$ yr$^{-1}$]. See Table S4 for classification of countries.

| Region | 2005 | 2025 | | | 2055 | | |
|---|---|---|---|---|---|---|---|
| | | SSP1 | SSP2 | SSP3 | SSP1 | SSP2 | SSP3 |
| Sub-Saharan Africa | 0 | 4 | 7 | 7 | 189 | 168 | 61 |
| Centrally-planned Asia and China | 0 | 0 | 0 | 0 | 0 | 0 | 0 |
| Central and Eastern Europe | 0 | 0 | 0 | 0 | 0 | 0 | 0 |
| Former Soviet Union | 0 | 18 | 22 | 32 | 13 | 42 | 54 |
| Latin America and the Caribbean | 0 | 1,157 | 1,441 | 1,715 | 975 | 1,846 | 2,896 |
| Middle East and North Africa | 2,389 | 2,454 | 3,353 | 3,689 | 15,245 | 37,610 | 44,890 |
| North America | 409 | 406 | 517 | 579 | 311 | 638 | 725 |
| Pacific OECD | 8 | 10 | 4 | 8 | 5 | 3 | 12 |
| Other Pacific Asia | 0 | 0 | 0 | 0 | 0 | 0 | 0 |
| South Asia | 0 | 0 | 0 | 0 | 1,964 | 0 | 0 |
| Western Europe | 0 | 0 | 0 | 0 | 0 | 0 | 0 |
| World | 2,806 | 4,050 | 5,344 | 6,030 | 18,702 | 40,306 | 48,638 |

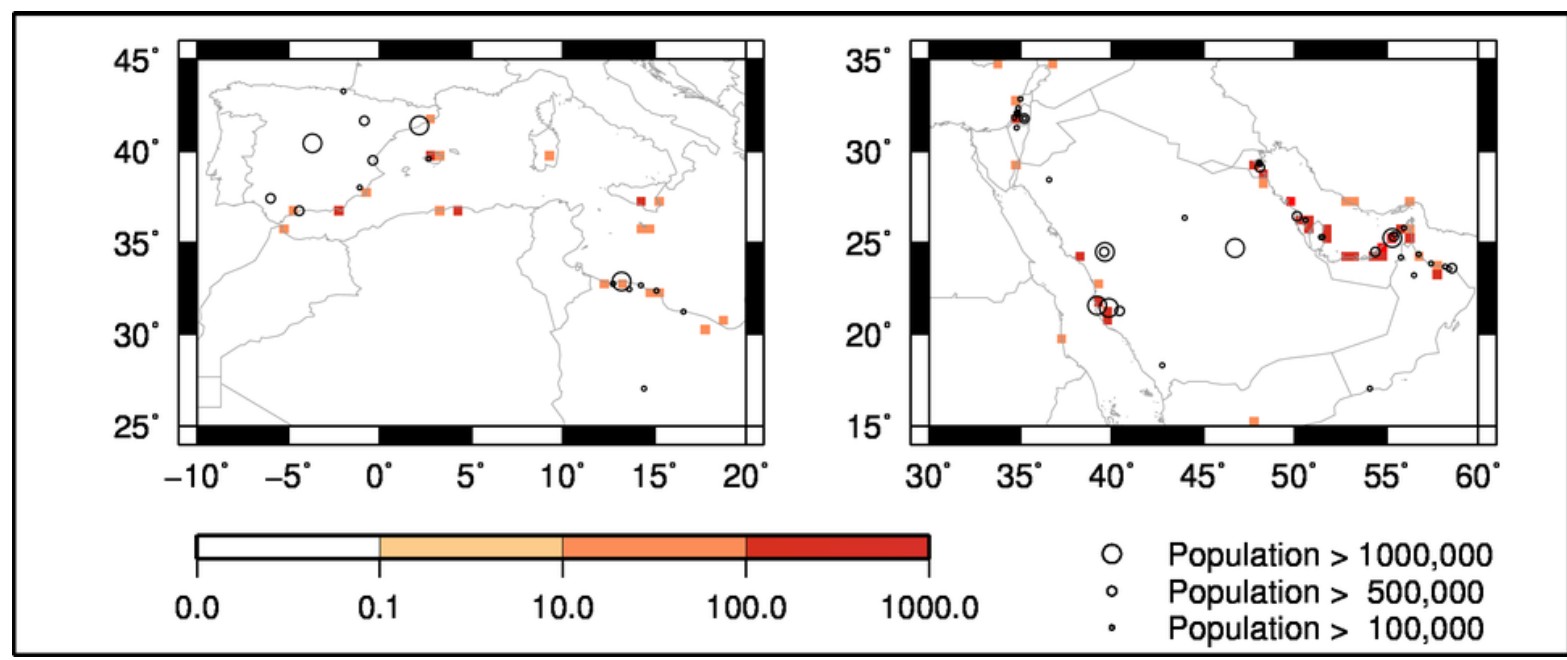

Figure 1 Actual locations of major desalination plants in the Mediterranean and Middle East in 2005. Boxes and circles represent plants (in 1000 m³ day⁻¹ of capacity; DesalData) and major cities (GeoNames), respectively.

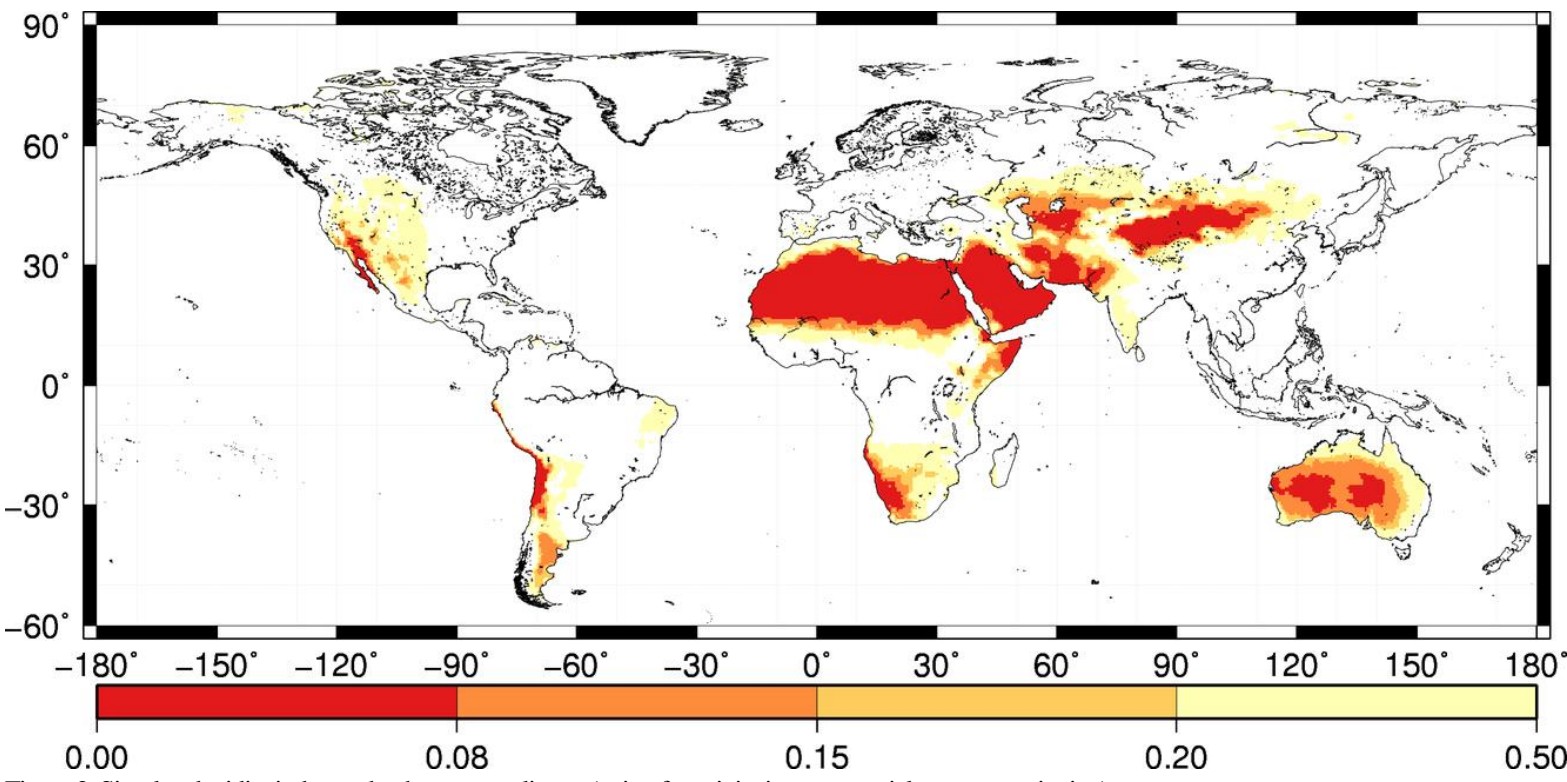

Figure 2. Simulated aridity index under the present climate (ratio of precipitation to potential evapotranspiration).

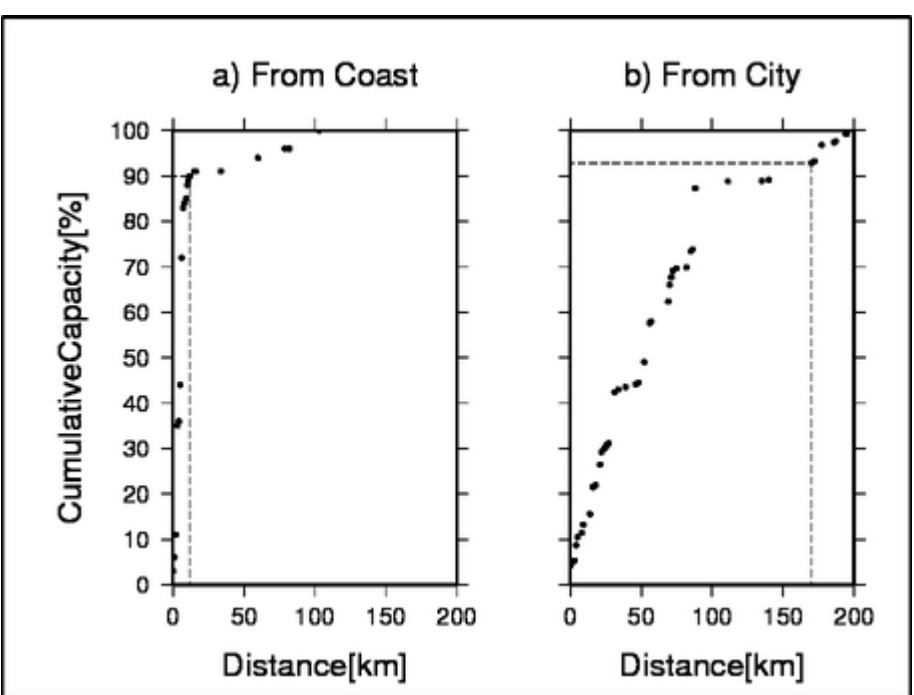

Figure 3. Cumulative capacity of desalination water and the distance of plants from the coastline (a) and major cities (b). The broken line shows the 90th percentile.

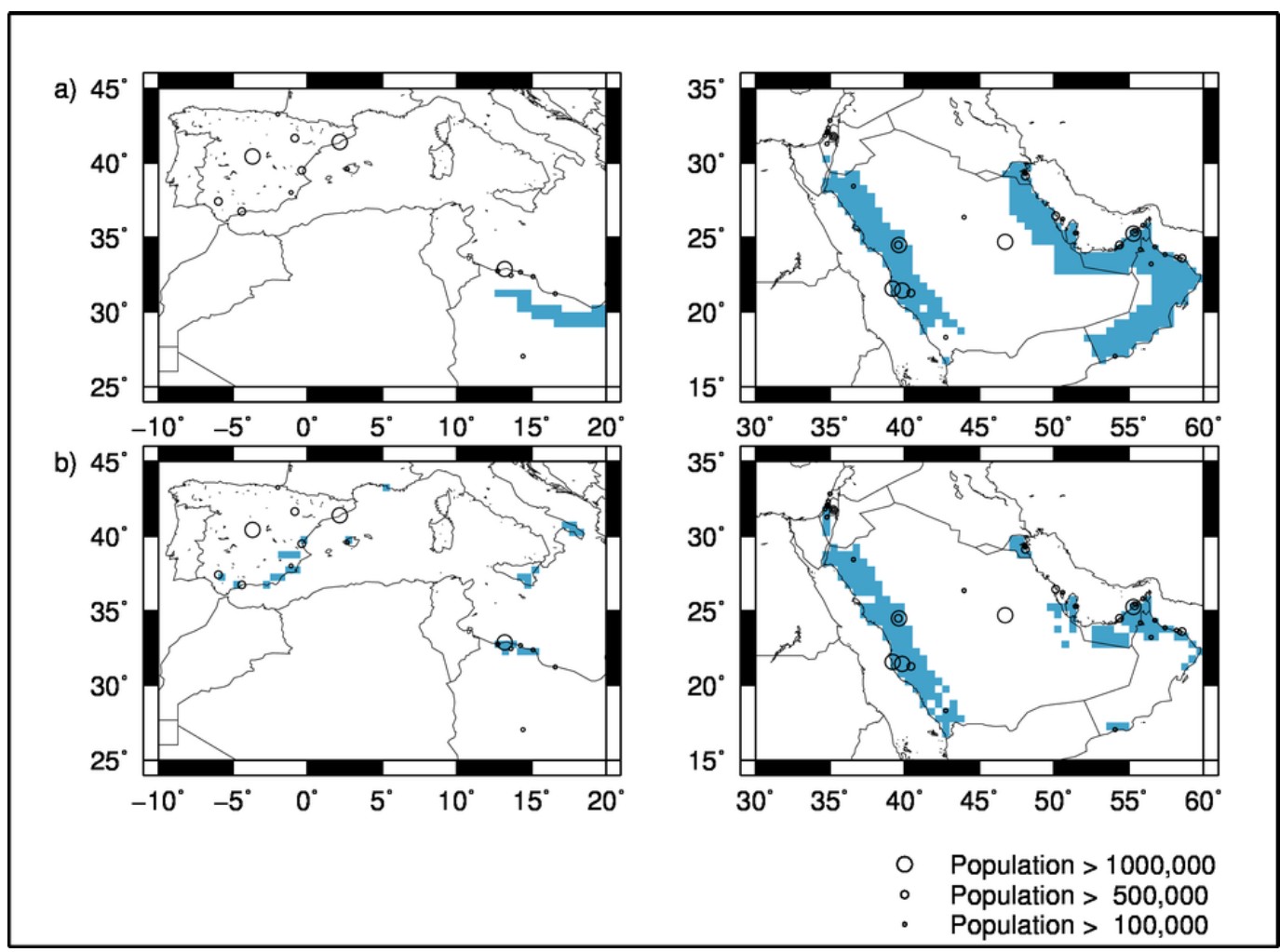

Figure 4. Simulated distribution of area utilizing seawater desalination (AUSD) in the Mediterranean and Middle East in 2005. a) Results from SDM, b) results from AUXSDM (see Section 3.2). Circles represent major cities (GeoNames).

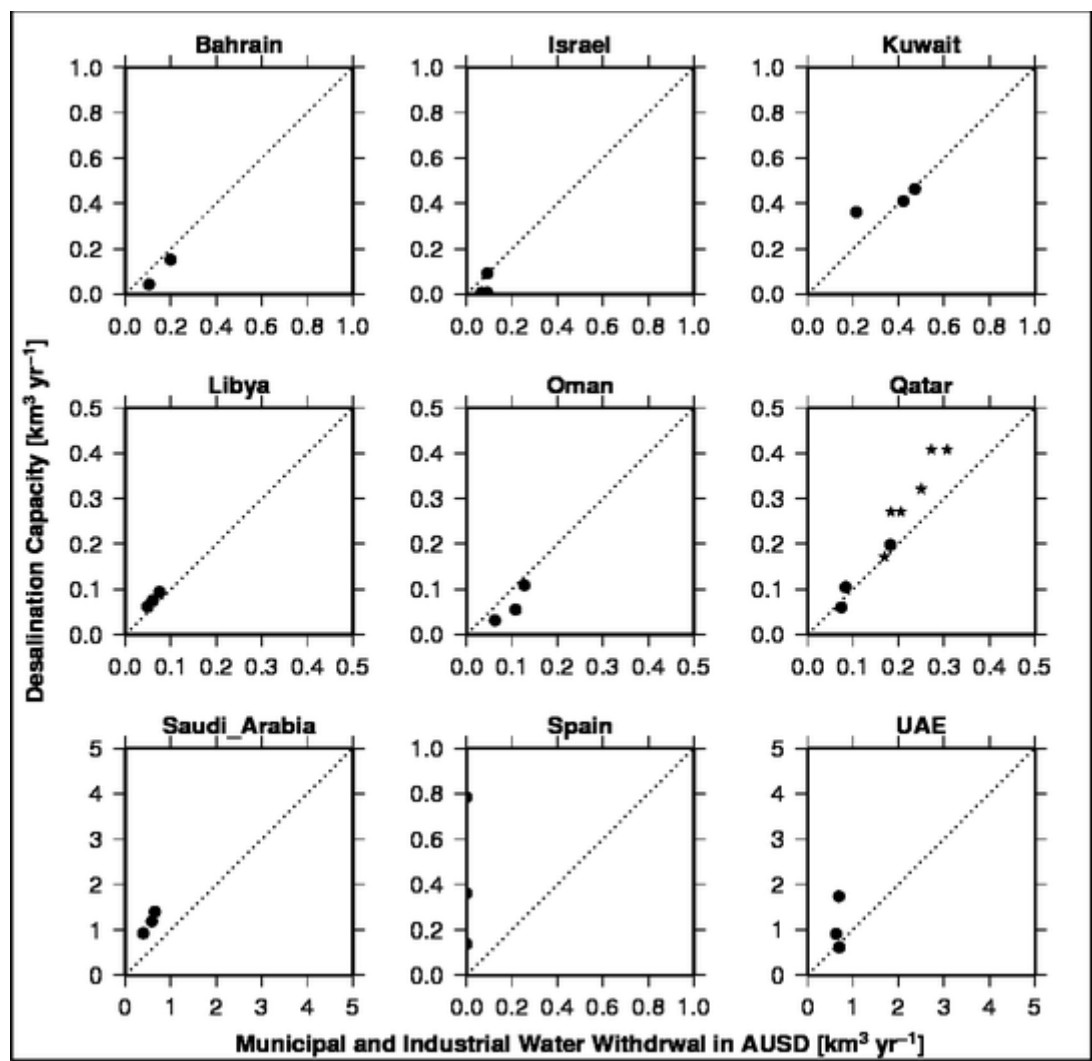

Figure 5. Comparison between national desalination capacity and municipal and industrial water withdrawal in AUSD. Each plot indicates one specific year. The shape of symbols indicate the data source of municipal and industrial water withdrawal in AUSD: circles for AQUASTAT and stars for MDPS (2016).

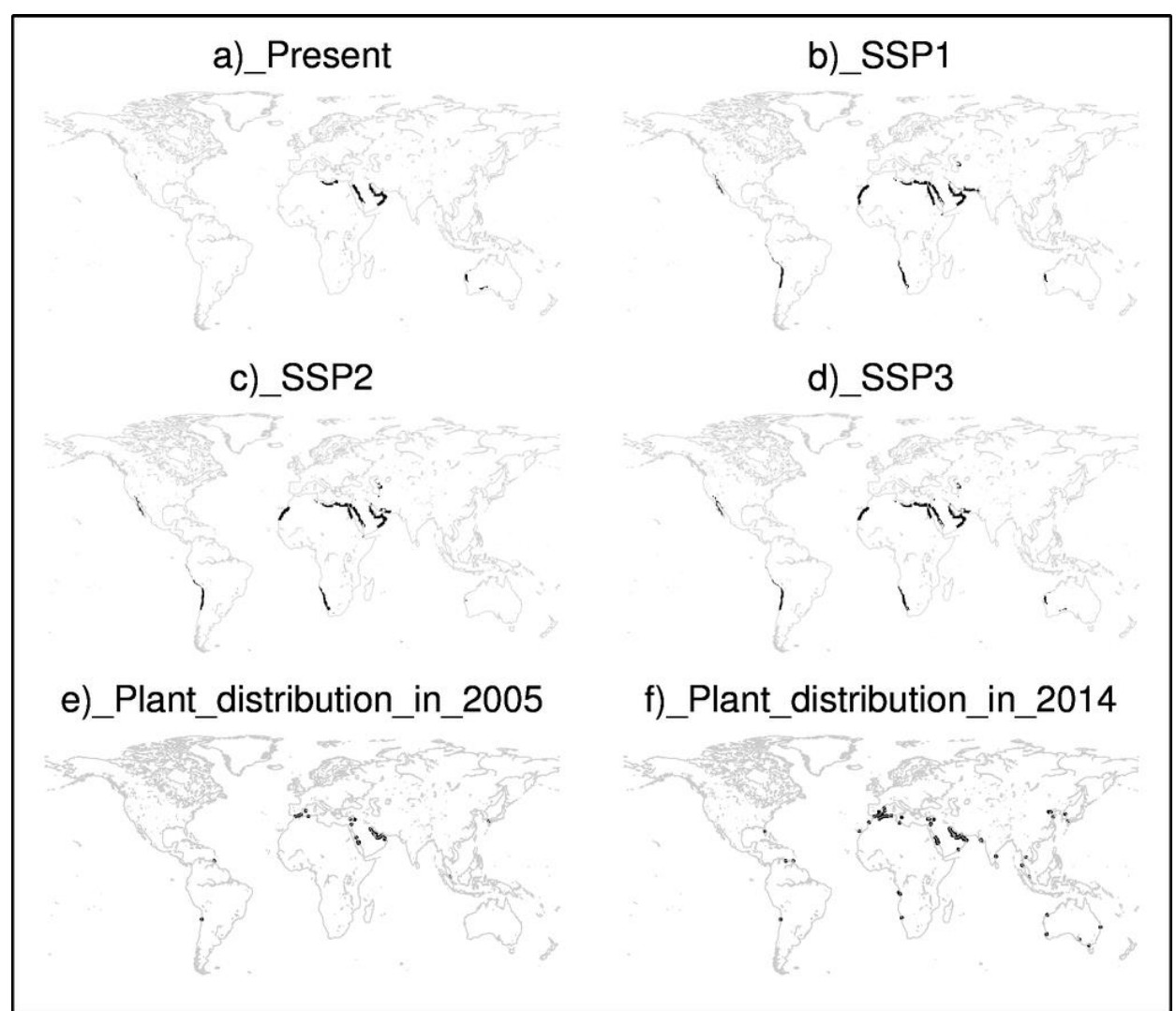

Figure 6. Global distributions of area utilizing seawater desalination (AUSD) in (a) 2005, (b) SSP1 in 2055, (c) SSP2 in 2055, and (d) SSP3 in 2055. Locations of seawater desalination plants (larger than 50,000 m$^3$ day$^{-1}$ in capacity) in (e) 2005 and (f) 2014.

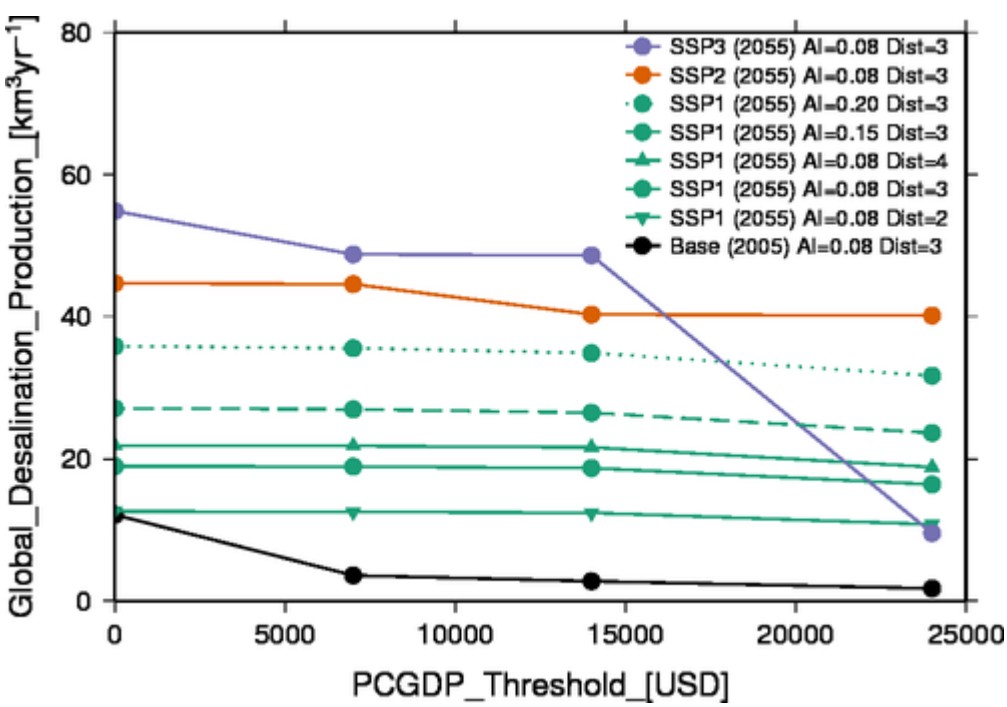

Figure 7. Sensitivity of per capita GDP (PCGDP; Condition A) and the aridity index (AI, Condition B) with respect to global total production of seawater desalination. Black, blue, green, and red lines for base year (2005), SSP1, SSP2, and SSP3 in 2055, respectively. Solid, broken, and dotted lines for AI = 0.08, AI = 0.15, and AI = 0.2, respectively. The sensitivity of AI is only shown for SSP1 in 2055.