# Peer review of "A seawater desalination scheme for global hydrological models"

_Hydrology and Earth System Sciences, 2016_

## Referee Comment (RC1) · Anonymous Referee #1 · 1 Apr 2016

The manuscript describes a new routine for incorporating seawater desalination dynamics in a global hydrologic model in this case the H08 model. The work is certainly an advancement over the current of the start of the art and how GHMs currently handle seawater desalination. So the work is of interest to the readership and would likely help advance the representations of desalination in GHMs. However, there are several major concerns especially with the proposed Seawater Desalination Model structure and the appropriateness of the validation exercise. I recommend major revisions.

Major concerns:

- The criteria for selecting desal plants (e.g, 613 from the 17,000 available in Desal-Data), the amendment of records as necessary such as in mountainous areas, and the selection of thresholds for the proposed conditions (e.g., 0.1 AI, 165km) seem ad

hoc and subjective, and irreproducible by others in case of the adjustments to the data. Also why was the model calibrated/tested for two countries and not all the countries for which desal exists? Also why not use procedures such as decision trees or regression decision trees, or the like to establish those thresholds based on the actual data as opposed to the guessing of thresholds?

- The validation of the SDM is inadequate. First, capturing the spatial distribution over a snapshot in time is an important element, but equally important is the ability of the model to simulate the evolution over time especially since the study considers desal water withdrawal over time as a central focus of the study. Second, the modeling approach is employing 3 conditions and assumptions as discussed in the paper to generate a gridded dependency map of areas dependent on desal plants globally with the goal to compare them to an observational dataset (DesalData) in this case. However, the only serious validation was a visual comparison between the results of Figure 4 and the desal plants location in two countries, Saudi Arabia and UAE. This visual comparison of the results (not even on the same figure and limited to two countries when the intent is to apply globally) is not sufficient to justify the appropriateness of the model. Also in figure 4, for example, the simulated areas/grids of area utilizing seawater desalination are widespread over much larger areal extent than the distribution of where the cities are located. For example, many of the inland grids between some of the major cities in the west of Saudi Arabia hardly have any population presence and a population based estimate might be more appropriate. Obviously many of these grids exhibit minimal water demands and where the major cities lie is what matters, but again this highlights the inadequacy of using figure 4 as the validation of the model. Also the comparison of country scale results is not sufficient to justify how the desal water withdrawals are distributed within a country. I would recommend expanding the validation exercise to global scale. For example, a scatter plot of total desal water use at the country scale for multiple time periods over history would help fill in this gap. Extending Figure 1 to show the global map and comparing that to Figure 5a would also be useful to show if the areas popping out as desal dependent over history are

consistent with where the desal plants are located around the globe.

- The first key question in the paper of looking at figuring out the common climatic and socioeconomic conditions associated with usage of seawater desalination around the global is a very interesting and important question, but as discussed in the previous two comments, the authors did not approach this in a systematic way and barely scratched off the surface in addressing this question. The paper is really mainly focused on the second question outlined by the authors.

- Why limit the analysis to large and seawater plants only? Also in many regions the use reverse osmosis and brackish water is more prevalent and in some cases many of the plants are actually inlands like in the US. Expanding to take a more holistic look at all of desal water use would make the paper a lot more useful to the scientific community. Minor concerns

Line 15: change 'likely' to 'projected'

Line 16: I would omit the term 'modern' throughout the manuscript when describing GHMs

Line 18: specify the historical period

Line 26: I would suggest to add a sentence or two at the end of the abstract to state the implications of those estimates (or provide a range of the fraction of local GDPs in countries that depend on desal or maybe translate the estimates to total energy instead of cost and compare to total energy consumption)

Line 80: 46.9 km3 sounds extremely large. Are you using the correct unit? It is confusing that units are km3/yr in the text and million m3/day in the tables. I would use consistently.

Line 84: it is not clear to why only Major Plants.

Line 85: '29.3' is the production or capacity? Also it would make the sentence more

informative to include the values for the other categories as well.

Line 91: 'erroneous' how were these circumstances adjusted. Clearly explain (maybe in SM) to ensure reproducibility of the work also who many data points were adjusted?

Line 105: add 'for each grid' before globally.

Line 110; I suspect there were both downscaled and bias-corrected. I would add a sentence to state so with a reference for completeness.

Line 127: to avoid having to send the readers to previous publications, I would suggest to consider reproducing these figures/table in the supplemental materials. Even better would be to tabulate the results at the country scale for the historical and future scenarios and placing them in an Excel file as a supplementary document.

Line 182: what does it mean to have 110

Line 213: why use 40-80 when the data says 28-77? How was that approximation done?

Line 269: I would show these numbers as fraction of regional or global GDPs. I would suggest doing this at the country scale at the least and giving a range.

Line 299: less than triple

Line 302: the increase of population from 14.5 to 184.3 million sounds really large! Is this due to the assumption of static population distribution within a country?

Table 1: what does other mean? I think adding a footnote to explain this sector and the value of 35 in Spain would be useful

Table 3: why 2055 in particular?

Table 4: several comments

- Replace 'historical' with 'historical (2005)'

- I would suggest using consistent units with those in the text (km3/yr).

- I would suggest adding or even replacing the last column on the right because it is hard to draw any conclusion from the total production cost. One option is to compute the range of the percent of GDP at the country scale to highlight that this maybe a substantial value in countries like Qatar or UAE for example. Another option is to look at the total energy associated with the production of that much desal and then compare to the total primary consumption in that country (in a way this would isolate the GDP portion not related to the energy sector). But the latter will like take some additional work so it may suffice to do the first option.

- Also consider adding the estimates from the study of Kim et al (2016). Balancing global water availability and use at basin scale in an integrated assessment model. Climatic Change, DOI 10.1007/s10584-016-1604-6

Figure 1: I would suggest adding 3rd figure at the bottom showing the whole globe and maybe showing all the plants excluded from the analysis in gray color. Also is color scale for capacity of production?

Figure 2: this is solely based on two countries and I would imagine that it should vary based on the size of the country and the distance from saline water bodies should matter in defining this threshold.

Figure 6: I would suggest omitting this figure and adding the results to table 4.

---

## Short Comment (SC1) · 1 Apr 2016

While this work has merit of enhancing the understanding of driving forces for desalination expansion, it needs substantial improvement to provide solid contribution to the community.

1. The methodology part is very vague, no model representations or equations are documented, and it is not clearly explicit so that the study can be reproduced.

2. The assumption of "All the municipal and industrial water withdrawal in Area Utilizing Seawater Desalination (AUSD) is supplied by seawater desalination" is unrealistic. Take the current biggest desalination country Saudi Arabia as an example, the surface freshwater withdrawal, groundwater withdrawal and desalinated water in 2005 is 1.1, 21.5 and 1.0 km3, respectively (AQUASTAT).

3. Only global total desalination amount for 2025 and 2055 is presented, the part that would be valuable to the community – the spatial and temporal changes of desalination amount – is missing.

---

## Referee Comment (RC2) · Anonymous Referee #2 · 2 Apr 2016

"A seawater desalination scheme for global hydrological models" by N. Hanasaki et al.

The paper presented by Hanasaki et al. developed a new modeling scheme for seawater desalination water use that can be applicable to existing global hydrological models. The newly developed desalinated water use scheme was then applied to project future desalinated water use under different socioeconomic (SSPs) and climate change (RCPs) scenario. It was found that future desalinated water use is expected to increase substantially (about 2-15 times), however, the future estimates vary significantly depending on different socioeconomic pathways. To my knowledge, this study provides most comprehensive results for historical and future desalinated water use estimates and projections across the globe. The author used the latest and very comprehensive data source, DesalData (http://desaldata.com/), and incorporated them into a global hydrological model. Existing global or large-scale studies on desalinated water use

typically use the data from FAO AQUASTAT, WRI EarthTrend, EuroStat, and available country statistics, which often have very limited global coverage. The authors combined the desalination data with other socioeconomic data (GDP, population, production cost, etc) to construct the Seawater Desalination Model (SDM). The paper is topical and presents interesting and useful findings, and it is concise and mostly well-written. The newly developed SDM is useful for large-scale modeling framework, and appears to be quite applicable to other GHMs. However, I do have some comments regarding the methodologies that were applied in SDM as detailed in the following.

1. The methodologies described in Section 2.4 is the key part of this study. The section is concise, however, it currently lacks rationales for those conditions and assumptions (A-C) made in the SDM. The methodologies appear to be quite arbitrary at its present form. Since this study focuses on developing a new desalinated water use model, these key parameters need to be described more thoroughly. I urge to expand Section 2.4 and provided further explanations of each key parameter, condition, and assumptions that have been incorporated in the SDM. These information are very useful for other large-scale hydrological modelers. Without further explanations, the novelty of this study is very limited. In addition, I suggest to combine Section 3.1 into Section 2.4. Section 3.1 is basically the method and background information that derived the key parameters.

2. Some assumptions made for the SDM are not entirely reasonable. For example, the cost of desalinated water use is decreasing and the efficiency of desalinated water use is improving in recent years. For future projections (towards 100 years later), further technological improvement is expected to reduce the cost drastically. The assumption made based on a historical trend may not be applicable for the future, e.g. desalinated water use for irrigation. In addition, fossil groundwater reserves are actively used in the Middle East and Northern Africa (MENA) despite the near zero natural recharge. The authors may include fossil groundwater use estimates to isolate the impact on desalinated water use. I would suggest to at least discuss these uncertainties further.

3. The uncertainty inherent in future water use estimates needs further discussion with some quantitative information (e.g., Wada et al., 2016).

Wada, Y., Flörke, M., Hanasaki, N., Eisner, S., Fischer, G., Tramberend, S., Satoh, Y., van Vliet, M. T. H., Yillia, P., Ringler, C., Burek, P., and Wiberg, D.: Modeling global water use for the 21st century: the Water Futures and Solutions (WFaS) initiative and its approaches, Geosci. Model Dev., 9, 175-222, doi:10.5194/gmd-9-175-2016, 2016.

4. I find the results presented are very interesting, and in particular, the substantial ranges in future desalinated water use estimates among different SSPs are intriguing. However, currently the paper focuses primarily on a global scale estimate but I think it is more beneficial to focus on regions and highlight the change in desalinated water use per country in MENA or other parts of the world. For example, the information like "As shown in Table 4, the volume of seawater desalination was estimated at 3.7 km3yr-1 with a cost of 1.5–14.0 109 USD, equivalent to 0.0025%–0.024% of total global GDP." is no so informative, in my opinion. This type of information should be provided at a country basis, since the regional heterogeneity of desalinated water use is extremely large.

5. Global figure like Figure 5 is not so informative, and I suggest to zoom in to some regions like Figure 4. This is a global scale study, but it should highlight more the regions of interest. The majority of the map is blank in Figure 5. The information density is very low.

6. Additional information on surface water availability per country (in some tables) for the future period would be useful to highlight the importance of desalinated water use. Water availability is generally projected to decrease over e.g., the Mediterranean.

7. I suggest the authors to make a similar table like Table 4 but focusing on some regions (Middle East, Mediterranean, etc). Table 4 is useful for a global comparison, but additional information on regional desalinated water use is also very useful including historical estimates and future projections like Table 1.

8. When you describe "14,000 USD", please be careful that this indicates per capita GDP only. Please check this throughout the manuscript.

9. The term "modern GHMs" is not clearly defined and not commonly used. This is rather confusing, what it is exactly indicating, e.g. model representations, framework, human components, etc. This should be corrected.

10. I think this study is important highlighting the significance of desalinated water use for coming decades. The paper can be a bit restructured with more regional focus.

In principal, I would recommend this paper to be considered for publication after some revisions.

---

## Editor Comment (EC1) · M. Sivapalan (Editor) · 27 May 2016

Dear Authors, as per HESS policy please post your responses to each of the reviewers' comments, including how you intend to revise the paper to address their concerns. In particular I would like you to engage with the reviewers in this way, which will only help you to improve the paper. Siva

———————————————————

---

## Author Comment (AC1) · 21 Jun 2016

Dear Editor,

The authors are grateful to you for taking time to handle our manuscript. We have received comments from two anonymous referees and Dr. Yaling Liu which were quite helpful to improve our paper. We have addressed all the comments and concerns as shown in the response to each. The manuscript has been thoroughly revised accordingly and ready to be submitted for possible consideration of another round of review.

Kind regards,

Naota Hanasaki

(on behalf of authors)

---

## Author Comment (AC2) · 21 Jun 2016

Dear Dr. Yaling Liu,

While this work has merit of enhancing the understanding of driving forces for desalination expansion, it needs substantial improvement to provide solid contribution to the community.

Thank you for taking time to read our discussion paper and provide valuable comments. Our response to your questions and comments are shown below.

1. The methodology part is very vague, no model representations or equations are documented, and it is not clearly explicit so that the study can be reproduced.

Thank you. Although we still believe all the necessary information is described in the discussion paper to reproduce our results, we further elaborated the details of methodology in the Method Section for broad readership. In short, our model overlays several gridded global maps of hydrological and socio-economic factors, and extract the grid cells that meets the conditions. Such fundamental concept has been now clearly shown in line 148-157 of the revised manuscript.

2. The assumption of "All the municipal and industrial water withdrawal in Area Utilizing Seawater Desalination (AUSD) is supplied by seawater desalination" is unrealistic. Take the current biggest desalination country Saudi Arabia as an example, the surface freshwater withdrawal, groundwater withdrawal and desalinated water in 2005 is 1.1,21.5 and 1.0 km3, respectively (AQUASTAT).

Thank you for giving us an opportunity to clarify some important points. The large volume of groundwater you pointed out is used for agriculture which is excluded from this study (we are focusing on only industrial and municipal water). In fact, AQUASTAT reports that as much as 88% ($20.83$ km$^3$ yr$^{-1}$) of total water withdrawal was used for agriculture in Saudi Arabia. Siebert et al. (2010) showed that 88.4% of irrigation water is supplied from groundwater in Arabian Peninsula. Hence the results shown in our discussion paper and the data provided by AQUASTAT are consistent.

3. Only global total desalination amount for 2025 and 2055 is presented, the part that would be valuable to the community – the spatial and temporal changes of desalination amount – is missing.

Thank you for this comment. In the revised manuscript, we have provided spatially and temporary detailed results. Spatially detailed results are now shown in newly added Table 5

(production of seawater desalination) and Tables S3-S4 (cost and its GDP percentage) and discussed in lines 310-330. Temporally (Historically) detailed results are shown in newly added Figure 5 and lines 281-294.

References

Siebert, S., Burke, J., Faures, J. M., Frenken, K., Hoogeveen, J., Döll, P., and Portmann, F. T.: Groundwater use for irrigation – a global inventory, Hydrol. Earth Syst. Sci., 14, 1863-1880, 10.5194hess-14-1863-2010, 2010.

---

## Author Comment (AC3) · 21 Jun 2016

Response to Reviewer 1

The manuscript describes a new routine for incorporating seawater desalination dynamics in a global hydrologic model in this case the H08 model. The work is certainly an advancement over the current of the start of the art and how GHMs currently handle seawater desalination. So the work is of interest to the readership and would likely help advance the representations of desalination in GHMs. However, there are several major concerns especially with the proposed Seawater Desalination Model structure and the appropriateness of the validation exercise. I recommend major revisions.

> Thank you very much for taking time to review and providing detailed and insightful comments to our work.
>
> Since we have received your comments, in order to fully respond to your questions and recommendations, we have collected additional data and evidences, improved models and algorithms, revisit the thresholds, and others. Owing to your comments, the model and paper have been largely improved. Although improved, the Seawater Desalination Model (SDM) is still not a perfect model. In this paper, we fully disclosed not only the strengths and detailed methods but also the limitations and shortcomings of SDM for possible future improvements by readers. We firmly believe that this study will be a first step to formulate desalination in global hydrological model.

Major concerns:
- The criteria for selecting desal plants (e.g, 613 from the 17,000 available in Desal-Data), the amendment of records as necessary such as in mountainous areas, and the selection of thresholds for the proposed conditions (e.g., 0.1 AI, 165km) seem ad hoc and subjective, and irreproducible by others in case of the adjustments to the data.

> Thank you. First, we added detailed rationale for selection of desalination plants in lines 15-36 of Supplemental Material. Second, about the amendments or records, now we added a list of desalination plants excluded from our analyses in Table S2 with reasons. Next, about thresholds, we agree with you that the thresholds in our model are dependent on the dataset used (e.g. Desal Data or others) and spatial resolution (e.g. 0.5°×0.5°grid cells or finer). We afraid, however, such dependency is common with all hydrological modeling (i.e. hydrological parameters are dependent on the selection of input meteorological data and spatial resolution). The readers who wish to reproduce our work would first compute AUSD with the threshold shown in our report, then adjust thresholds if necessary by comparing it with the actual distribution of desalination plants and volume of water production. These

procedures are now added lines 167-168.

Also why was the model calibrated/tested for two countries and not all the countries for which desal exists?

> First, we validated our model for nine countries (see Table 1 and lines 204-209), not two countries. These nine countries produces 85% of seawater desalination in the world in 2005. We further elaborated these two countries in text because detailed municipality-specific statistics were obtained (Table 2). We tried to find similar statistics for other countries, but it was not successful. Note that collecting data for water production and use is quite difficult partly due to language: we have been searching documents written in English, not in local languages. These limitations are now added in Discussion (lines 435-439).

Also why not use procedures such as decision trees or regression decision trees, or the like to establish those thresholds based on the actual data as opposed to the guessing of thresholds?

> Thank you for this suggestion. Following your advice, first we estimated the thresholds in SDM by the decision tree method. The thresholds obtained didn't reproduced the reality because they were erroneously influenced by data outliers. To avoid this problem, we also tested the random forest method. The results were improved, but due to considerable difference among the iteration runs, we could not determine a single value for each threshold. Finally, we adopted the segmented regression method to estimate the threshold of aridity index (AI) which was determined by visual inspection in the discussion paper. We plotted the relationship between the AI of plants located and their capacity. It shows remarkable difference in the distribution of plots by AI. We applied the segmented regression method and statistically estimated the break point at 0.075 and set as the new threshold (original number shown in the discussion paper was 0.1). We have re-calculated all the numbers using this new threshold and revised throughout the text. The revised methods are described in lines 214-217.

- The validation of the SDM is inadequate. First, capturing the spatial distribution over a snapshot in time is an important element, but equally important is the ability of the model to simulate the evolution over time especially since the study considers desalwater withdrawal over time as a central focus of the study.

> Thank you for this comment. It was not possible to conduct a retrospective simulation

because of lack of key input data (i.e. GDP per capita and water withdrawal for industrial and municipal purposes). Therefore, we added a new figure (Figure 5) and text (lines 281-294) showing the country-base historical variations in desalination capacity and water withdrawal. The results showed that the temporal change in domestic and industrial water withdrawal in AUSD corresponds fairly well with that of the total capacity of seawater desalination, which promises the temporal evolution in simulation.

Second, the modeling approach is employing 3 conditions and assumptions as discussed in the paper to generate a gridded dependency map of areas dependent on desal plants globally with the goal to compare them to an observational dataset (DesalData) in this case. However, the only serious validation was a visual comparison between the results of Figure 4 and the desal plants location in two countries, Saudi Arabia and UAE. This visual comparison of the results (not even on the same figure and limited to two countries when the intent is to apply globally) is not sufficient to justify the appropriateness of the model.

Thank you very much for this comment. Actually, SDM was validated from three perspectives: geographical extent of AUSD (Figure 4, lines 248-268), total desalination water production in 2005 (Table 1 and Figure S2, lines 269-280), and temporal change in production (Figure 5; newly added in revisied manuscript; lines 281-294). In order to make this structure clearer, now we added a summary of validation in lines 295-300.

Also in figure 4, for example, the simulated areas/grids of area utilizing seawater desalination are widespread over much larger areal extent than the distribution of where the cities are located. For example, many of the inland grids between some of the major cities in the west of Saudi Arabia hardly have any population presence and a population based estimate might be more appropriate. Obviously many of these grids exhibit minimal water demands and where the major cities lie is what matters, but again this highlights the inadequacy of using figure 4 as the validation of the model.

Thank you for this comment and sorry if our results confused you. As you pointed out, the areal extent of AUSD and the locations of individual desalination plants are not directly comparable. AUSD is expected to include major desalination plants, but it is not necessarily true that all of the AUSD grid cells contain major desalination plants. The definition and interpretation of AUSD is thoroughly revised and shown in lines 148-157. Additionally, we revised the whole text not to confuse readers.

Also the comparison of country scale results is not sufficient to justify how the desal water withdrawals

are distributed within a country. I would recommend expanding the validation exercise to global scale. For example, a scatter plot of total desal water use at the country scale for multiple time periods over history would help fill in this gap. Extending Figure 1 to show the global map and comparing that to Figure 5a would also be useful to show if the areas popping out as desal dependent over history are consistent with where the desal plants are located around the globe.

> Thank you for an excellent idea. Now we added two new panels to Figure 6 (Figure 5 in the original discussion paper), showing the location of major desalination plants globally in 2005 and 2014 (latest data accessible). It shows, for instance, AUSD in 2055 expands into coastal Chile and southwestern Africa, where some major desalination plants actually exist . In terms of temporal dynamics, we newly added Figure 5 showing the scatter plot for multiple time periods over history.

- The first key question in the paper of looking at figuring out the common climatic and socioeconomic conditions associated with usage of seawater desalination around the global is a very interesting and important question, but as discussed in the previous two comments, the authors did not approach this in a systematic way and barely scratched off the surface in addressing this question. The paper is really mainly focused on the second question outlined by the authors.

> Thank you. As you pointed our research question was a bit too broad in contrast to what we obtained so far and data availability. Now we have rephrased the first question to make consistent with what we have actually done, which is not discovering "the common condition", but investigating some characteristics in present desalination plants (lines 69-72).

- Why limit the analysis to large and seawater plants only? Also in many regions the use reverse osmosis and brackish water is more prevalent and in some cases many of the plants are actually inlands like in the US. Expanding to take a more holistic look at all of desal water use would make the paper a lot more useful to the scientific community.

> Thank you. To develop a general model on desalination plant would be very useful for larger community, but highly challenging. Particularly, inland desalination is of a matter of not only available water quantity but required water quality which is difficult to simulate with global models. As we stated in Introduction (lines 67-75), in this study, we focused on the role of seawater desalination that sustains local water usage.

Minor concerns

Line 15: change 'likely' to 'projected'

Line 16: I would omit the term 'modern' throughout the manuscript when describing GHMs

Line 18: specify the historical period

Line 26: I would suggest to add a sentence or two at the end of the abstract to state the implications of those estimates (or provide a range of the fraction of local GDPs in countries that depend on desal or maybe translate the estimates to total energy instead of cost and compare to total energy consumption)

      Thank you. We corrected all.

Line 80: 46.9 km3 sounds extremely large. Are you using the correct unit? It is confusing that units are km3/yr in the text and million m3/day in the tables. I would use consistently.

      Thank you. As of June 2014, DesalData included records for 17423 desalination plants globally. The number 46.9km$^3$ was the simple summation of capacity of all the plants recorded. We noticed that the number includes plants in "offline" and "planned" status. In the revised text, we calculated the total capacity of online plants only which amounts 21.9 km$^3$.

Line 84: it is not clear to why only Major Plants.

      Thank you. We added detailed rationale of plant selection criteria in Supplemental Text (lines 15-36).

Line 85: '29.3' is the production or capacity? Also it would make the sentence more informative to include the values for the other categories as well.

      Thank you for pointing out this. For the same reason stated above, now we only summed up the plants in "online" status. Now it reads 13.3 km$^3$ yr$^{-1}$. Since general seawater desalination information is available elsewhere (e.g. Voutchkov, 2012), in order to keep our text concise, we are afraid but we didn't include numbers for other categories.

Line 91: 'erroneous' how were these circumstances adjusted. Clearly explain (maybe in SM) to ensure reproducibility of the work also who many data points were adjusted?

Thank you. We listed eleven plants excluded with reasons in Table S2.

Line 105: add 'for each grid' before globally.

Thank you, we added.

Line 110; I suspect there were both downscaled and bias-corrected. I would add a sentence to state so with a reference for completeness.

Thank you. We mentioned that the methods of downscaling and bias correction are identical to those introduced in Hanasaki et al. (2013a,b)

Line 127: to avoid having to send the readers to previous publications, I would suggest to consider reproducing these figures/table in the supplemental materials. Even better would be to tabulate the results at the country scale for the historical and future scenarios and placing them in an Excel file as a supplementary document.

Thank you for this suggestion. Perhaps most important information for the readers is not figures but the assumptions of future scenarios, which is only available in the original references. Because the references are both published in an open access journal, it doesn't take much time to view them, we believe.

Line 182: what does it mean to have 110

Thank you. As shown in the next sentence, it is due to inconsistency in data. (line 209).

Line 213: why use 40-80 when the data says 28-77? How was that approximation done?

Thank you. We agree with you. The lower boundary of 40% is not justified from the data. Based on the minimum value appeared in Table 2, we modified the threshold as 30-80%.

Line 269: I would show these numbers as fraction of regional or global GDPs. I would suggest doing this at the country scale at the least and giving a range.

Thank you. We added regional numbers as fraction of regional GDP in Table S4. Taking into account the number of assumptions and reproducibility of the past, we believe that the

model is not in the stage for country-base assessment.

Line 299: less than triple

Thank you. We corrected it and now it reads "more than double".

Line 302: the increase of population from 14.5 to 184.3 million sounds really large! Is this due to the assumption of static population distribution within a country?

Thank you. This is mainly due to expansion of AUSD in North Africa (for all scenarios) and South Asia (only for SSP1; see Figure 6).Particularly, when densely populated Egypt and Pakistan becomes AUSD, the population living in AUSD showed a rapid increase (line 327-332).

Table 1: what does other mean? I think adding a footnote to explain this sector and the value of 35 in Spain would be useful

Thank you for this comment. We added a footnote and noticed that this is used for irrigation.

Table 3: why 2055 in particular?

Thank you. We have added numbers for 2005 and 2025, because we intensively analyzed these periods as well.

Table 4: several comments

- Replace 'historical' with 'historical (2005)'

Because some earlier studies didn't report the year of 2005, we kept as is.

- I would suggest using consistent units with those in the text (km3/yr).

Thank you. We used the unit of $10^6$ m$^3$ yr$^{-1}$

- I would suggest adding or even replacing the last column on the right because it is hard to draw any conclusion from the total production cost. One option is to compute the range of the percent of GDP

at the country scale to highlight that this maybe a substantial value in countries like Qatar or UAE for example. Another option is to look at the total energy associated with the production of that much desal and then compare to the total primary consumption in that country (in a way this would isolate the GDP portion not related to the energy sector). But the latter will like take some additional work so it may suffice to do the first option.

> Thank you very much for this suggestion. We added percent GDP of seawater desalination cost of the world in Table 4. The same indicator are now reported in Table S4 for eleven regions. Again, since too many assumptions and uncertainties were included in the model and settings, country-base results were not shown in the revised text.

- Also consider adding the estimates from the study of Kim et al (2016). Balancing global water availability and use at basin scale in an integrated assessment model. Climatic Change, DOI 10.1007/s10584-016-1604-6

> Thank you for letting us know the work of Kim et al. (2016). We mentioned this interesting work in Introduction. Since quite limited quantitative results were provided (only the total desalination production in 2100), we couldn't include their results in Table 4.

Figure 1: I would suggest adding 3rd figure at the bottom showing the whole globe and maybe showing all the plants excluded from the analysis in gray color. Also is color scale for capacity of production?

> We displayed the present location of major desalination plants in Figure 6. The color scale of Figure 1 shows the capacity of production.

Figure 2: this is solely based on two countries and I would imagine that it should vary based on the size of the country and the distance from saline water bodies should matter in defining this threshold.

> About the first point, actually Figure 3 is based on the data for nine Major Countries (please see Figure 1, red cells extends in many countries). Hence we believe that the results are not dependent on the national characteristics of Saudi Arabia and UAE. We added this point in line 218-219.

Figure 6: I would suggest omitting this figure and adding the results to table 4.

> Thank you for suggestion. We have tried this, but it made Table 4 very complex. We kept

Figure 6 (Figure 7 in revised manuscript) as is.

---

## Author Comment (AC4) · 21 Jun 2016

Response to Reviewer 2

The paper presented by Hanasaki et al. developed a new modeling scheme for seawater desalination water use that can be applicable to existing global hydrological models. The newly developed desalinated water use scheme was then applied to project future desalinated water use under different socioeconomic (SSPs) and climate change (RCPs) scenario. It was found that future desalinated water use is expected to increase substantially (about 2-15 times), however, the future estimates vary significantly depending on different socioeconomic pathways. To my knowledge, this study provides most comprehensive results for historical and future desalinated water use estimates and projections across the globe. The author used the latest and very comprehensive data source, DesalData (http://desaldata.com/), and incorporated them into a global hydrological model. Existing global or large-scale studies on desalinated water usetypically use the data from FAO AQUASTAT, WRI EarthTrend, EuroStat, and available country statistics, which often have very limited global coverage. The authors combined the desalination data with other socioeconomic data (GDP, population, production cost, etc) to construct the Seawater Desalination Model (SDM). The paper is topical and presents interesting and useful findings, and it is concise and mostly wellwritten. The newly developed SDM is useful for large-scale modeling framework, and appears to be quite applicable to other GHMs. However, I do have some comments regarding the methodologies that were applied in SDM as detailed in the following.

Thank you very much for taking time to review and providing insightful comments. All of the concerns you raised have now addressed as below.

1. The methodologies described in Section 2.4 is the key part of this study. The section is concise, however, it currently lacks rationales for those conditions and assumptions (A-C) made in the SDM. The methodologies appear to be quite arbitrary at its present form. Since this study focuses on developing a new desalinated water use model, these key parameters need to be described more thoroughly. I urge to expand Section 2.4 and provided further explanations of each key parameter, condition, and assumptions that have been incorporated in the SDM. These information are very useful for other large-scale hydrological modelers. Without further explanations, the novelty of this study is very limited. In addition, I suggest to combine Section 3.1 into Section 2.4. Section 3.1 is basically the method and background information that derived the key parameters.

Thank you for this comment. Now Section 2.4 is further elaborated. First, taking the

comments and questions of Reviewers 1 and 2, we have elaborated the concept of SDM and parameters (lines 148-157). In the original discussion paper, the parameters (thresholds in Conditions A-C and Assumptions A-C) are suddenly appeared without sufficient explanation. Now it is clearly described how the parameters were derived (lines 165-171 and 177-186). Because many reviewers and readers like to distinguish methods and the results of data analyses, we kept the subsections 2.4 and 3.1 as is, but we further edited for readability.

2. Some assumptions made for the SDM are not entirely reasonable. For example, the cost of desalinated water use is decreasing and the efficiency of desalinated water use is improving in recent years. For future projections (towards 100 years later), further technological improvement is expected to reduce the cost drastically. The assumption made based on a historical trend may not be applicable for the future, e.g. desalinated water use for irrigation. In addition, fossil groundwater reserves are actively used in the Middle East and Northern Africa (MENA) despite the near zero natural recharge. The authors may include fossil groundwater use estimates to isolate the impact on desalinated water use. I would suggest to at least discuss these uncertainties further.

Thank you. We completely agree with you that technology advances and subsequent cost reductions will enhance the introduction of desalination plants. We noted that this mechanism is not included in SDM in lines 428-432. Note that importance of this mechanism and possible distortion of simulation results have been already discussed elsewhere (lines 386-390). Also, this mechanism have been partly analyzed in the sensitivity test of changing the threshold of Condition A (minimum per capita GDP; Figure 7): the decrease in the threshold corresponds to diminish in the relative cost of desalination. We also fully notice the importance of alternative water sources. As you pointed out, the production of desalination is largely determined by the availability of renewable/fossil groundwater. This could be partly achievable if we combine SDM with the global hydrological model that explicitly simulate renewable/fossil groundwater. We added discussion in lines 432-434.

3. The uncertainty inherent in future water use estimates needs further discussion with some quantitative information (e.g., Wada et al., 2016).
Wada, Y., Flörke, M., Hanasaki, N., Eisner, S., Fischer, G., Tramberend, S., Satoh, Y., van Vliet, M. T. H., Yillia, P., Ringler, C., Burek, P., and Wiberg, D.: Modeling global water use for the 21st century: the Water Futures and Solutions (WFaS) initiative and its approaches, Geosci. Model Dev., 9, 175-222, doi:10.5194/gmd-9-175-2016, 2016.

Thank you. Wada et al. (2016) is particularly relevant to Section 3.5 "Key uncertainties and implications". We cited this paper and mentioned that the water use projections substantially differ among models and systematic model inter-comparison is under way (lines 411-412).

4. I find the results presented are very interesting, and in particular, the substantial ranges in future desalinated water use estimates among different SSPs are intriguing. However, currently the paper focuses primarily on a global scale estimate but I think it is more beneficial to focus on regions and highlight the change in desalinated water use per country in MENA or other parts of the world. For example, the information like "As shown in Table 4, the volume of seawater desalination was estimated at 3.7 $km^3yr^{-1}$ with a cost of 1.5–14.0 109 USD, equivalent to 0.0025%–0.024% of total global GDP." is no so informative, in my opinion. This type of information should be provided at a country basis, since the regional heterogeneity of desalinated water use is extremely large.

In the revised manuscript, we showed the change in the volume of seawater desalination for 11 regions in the world in terms of desalinated water production (Table 5) and cost (Table S3 and S4). Discussion is shown in lines 317-337.

5. Global figure like Figure 5 is not so informative, and I suggest to zoom in to some regions like Figure 4. This is a global scale study, but it should highlight more the regions of interest. The majority of the map is blank in Figure 5. The information density is very low.

Thank you for your suggestions. We tried your idea but it was not very successful. Due to income growth by the mid of 21$^{st}$ century, regions including AUSD increases substantially in all continents which makes the figure highly complex.

6. Additional information on surface water availability per country (in some tables) for the future period would be useful to highlight the importance of desalinated water use. Water availability is generally projected to decrease over e.g., the Mediterranean.

Thank you. This is an important point. Now we have added discussion in lines 379-381.

7. I suggest the authors to make a similar table like Table 4 but focusing on some regions (Middle East, Mediterranean, etc). Table 4 is useful for a global comparison, but additional information on regional desalinated water use is also very useful including historical estimates and future projections like Table

1.

> Thank you for this suggestion. As stated above, regional projections are now shown in Tables 5, S3, and S4. Discussion is shown in lines 317-337.

8. When you describe "14,000 USD", please be careful that this indicates per capita GDP only. Please check this throughout the manuscript.

> Thank you for noticing this. We have checked the unit of per capita GDP throughout text.

9. The term "modern GHMs" is not clearly defined and not commonly used. This is rather confusing, what it is exactly indicating, e.g. model representations, framework, human components, etc. This should be corrected.

> Thank you. We've also received similar comment from another reviewer. Now we removed adjective "modern" from text. The original intention was global hydrological models including latest treatment of human activities, but we agree that the adjective was simply confusing.

10. I think this study is important highlighting the significance of desalinated water use for coming decades. The paper can be a bit restructured with more regional focus. In principal, I would recommend this paper to be considered for publication after some revisions.

> Thank you. Now we have revised our paper trying to include your suggestions as much as possible. We believe now the paper has been substantially improved including extended discussion on regional results.

---

## Author Response (AR2)

Dear Editor,

Thank you very much for taking the time to consider our paper entitled "A seawater desalination scheme for global hydrological models." We are grateful to all of the reviewers for providing detailed and insightful comments. We have responded to all of the suggestions and concerns as shown below.

Reviewer 1

Although the authors have tried to address the concerns that were raised during the first round of review, I am still not satisfied with the quality and the scientific contribution of this paper to warrant its publication in HESS.

> We are grateful to Reviewer 1 for the insightful comments, which we took into serious consideration to improve our paper. This study presents the first model that quantitatively estimates the geographical distribution and volume of seawater desalination production. We firmly believe that this paper provides an important first step for this challenging and emerging research field, in noting that the performance of our model is in certain circumstances below the standard of traditional hydrological models dealing with the behaviors of well monitored natural catchments. We emphasize that the first development of all models (e.g. global climate models, land surface models) begins with a simple version, typically limited in function and performance compared with later iterations. The shortcomings of our model are discussed, which we believe is a critical first step towards improving it in future studies, by ourselves and by the greater scientific community.

In a nutshell, the paper relies on a large dataset of desalination plants (DesalData) to identify grid cells that rely on desalinated water through a set of predetermined conditions (GDPC>14k, aridity<7.5%, distance<~165km), and then through a set of assumptions, the model establishes the amount of desalinated water that would need to be produced. The established thresholds are kept the same globally and over time and as such they could yield strong implications about the future dependence on desalinated water. For example, the per capita income threshold of 14k does not account for the depreciation value of money over time, and since GDPC is expected to increase under all the 5 SSPs, this assumption will lead to wide spread of desal under all the SSP scenarios even the defragmented (and relatively poor) world of SSP3.

We are grateful for these comments and the opportunity to clarify some technical issues. GDP per capita is in 2005 USD Purchasing Power Parity units. Both monetary depreciation over time and difference in prices among nations/regions have been excluded from GDP projections, as noted in lines 146-148. In addition, the increased prevalence of desalination under all SSP scenarios is primarily attributed to a marked increase in GDP for all of the nations, even after removing the effects of depreciation. Please note that the world of SSP3 is "poorer" than the other four scenarios, but it is apparently "richer" under future scenarios than today. In general, we agree that the world economic view of SSP is quite optimistic, but that discussing the validity of the SSP scenario is beyond the scope of this paper. We noted this point in lines 400-404.

As ground water resources start to deplete, there is no guarantee that the 165km threshold will hold and this is already seen in the case of the city of Riyadh, which is about 400km from the nearest coast. The same can be said about the aridity index threshold.

We agree that future model thresholds may change, due to alterations in water availability, technological change, water use, and other factors. Such factors are hard to quantify and include in our model. What is possible is to conduct sensitivity tests, and add the obtained results to the discussion. As such, a sensitivity test on aridity index (Figure 7) has already been conducted, and we added a new test concerning distance from the nearest coast (lines 441-443). We have also expanded the discussion of potential limitations arising from future changes in thresholds in lines 488-492 and 527-530.

Also the fact that conditions/threshold/assumptions were based on the same data that the authors are using to validate the model defeats that purpose. One would expect there is an independent dataset for calibration and another one for validation, which was not the case for this study. Even when basing the validation on the actual year (2005) to derive the grid areas depending on desalinated water and their production estimates, there are many exceptions as discussed by the authors (e.g., Spain and Israel).

First, it is quite difficult to take a systematic and standard approach towards model calibration and validation due to the patchy availability of data concerning desalination and water use. As such, we carried out simple tests to confirm the robustness of the parameters. We randomly split the data and estimated the three most important thresholds (aridity and distance from coastlines and cities) for each split. These results are close to the original estimates of 0.082, 15, and 170.

Second, we developed a new model (SDM2) and reported key results in the revised paper. The SDM2 estimates the regions in which desalination water is used in supplementary way. This model substantially improved the distribution of Area Utilizing Seawater Desalination (AUSD) in the Mediterranean region, including in Spain and Israel (compare Figure 1 and Figure 4b). Model description and results are shown in lines 272-284 and lines 310-323 respectively. Please also see our response to Reviewer 3 which includes more detailed discussion on SDM2.

Finally, the paper can still improve from proofreading to clarify the writing. I have tried to highlight some examples below.

Thank you for this comment. The revised version has undergone proofreading by two professional editors, whose first language is English.

Line 18: change 'period' to 'year'

Thank you for this comment. This is corrected in the revised version.

Lines 25-26: I would suggest omitting the second to last sentence. If you keep it, I would reduce the number of significant figures used or use billion $ unit instead of millions (1.1-10.5 & 1.5-22.1 Billion USD). Although I am not familiar with how much desal plants typically cost, but the numbers do look small. Is it only the capital investment cost that you have accounted for? or also the levelized cost over the lifetime of the plant including operation and maintenance, and the energy cost often associated with such activities, plus piping structures, pumping and distribution costs?

Thank you for these points of clarification. These costs are now reported in billion USD, with one decimal place included. Secondly, the costs now include both capital investment and operational costs. We then rounded the total cost by assuming a plant life of 20 years and a discount rate of 8%, as per Lamei et al. (2008). A brief method summary is also now included in the Supplemental Text lines S62–S72.

Line 32: 'there is a growing concern over the sustainability of'

Thank you. This has now been corrected in the revised draft.

Line 45: Add a more recent citation beside (Bremere et al., 2001).

Following a search of the literature, we have now included a citation of a report by Fichtner GmbH, a private consultancy firm, published in 2011. This includes a summary of the report (lines 65-69), its projections (Table 4), and a discussion of its results (lines 392-398).

Line 47: the introduction lacks any information about the current capacity & production of desal and even the historical evolution beyond citing figure S1. For example, when citing that Kim et al came up with the estimate of 250km3/yr by 2100, it is hard for the reader to give sense to how big or small that number is in comparison to current use.

Thank you. We added the total production of desalinated water in 2005 (5.2 km$^3$ yr$^{-1}$).

Line 91: give the numbers for the rest of categories, 'followed by brackish water, saline inland and river water, and others.' If I am understanding this correctly, the 13.3 is the largest portion of the 21.9 total, and the other categories add up to the remaining 8.6km3, right?

Yes, this is correct.

Line 93: "First, we selected the desalination plants (hereafter Major Plants) to be included in our analyses." Are not you missing the word 'major' in that sentence?

Thank you for catching this; the word 'major' is now included.

Lines 93-94: "The detailed selection criteria and the rationale is shown in Table S1 and Supplemental Text." Table S1 lists the criteria but does not say anything about the rationale. For example, it is still not clear what was the rationale for using only 613 large of 17000+ plants available? The SM does not really address this point as claimed by the authors. The only piece that relates to this aspect is the following "Finally, we selected plants of larger than 10000m3d-1 of capacity. It corresponds to approximately 1.5mm yr-1 of water resources assuming the area of a grid cell is 2500km2(0.5°×0.5°), which is same order of the magnitude of other water resources components simulated in H08."

As the Supplemental Text of the original version includes a rationale for using a threshold of 10,000 m$^3$ d$^{-1}$, we request clarification about the additional information requested. Given that questions regarding the number of plants were included within the comments, we responded by adding Table S3 in the Supplemental Material, which shows the number of plants and total desalination capacity under various conditions. We used 559 plants in our

analysis (excluding 54 additional records, including erroneous longitude and latitude). This covers a considerable portion of the total desalination capacity globally (81% of the total capacity of online plants of all sizes using seawater). Thus, we argue that the exclusion of small plants from our analysis would not have major effects, given their relatively small contribution. Text regarding this issue was added to the discussion (lines 117-121).

Lines 95-96: "For example, DesalData contains some records indicating that plants in mountainous inland regions use seawater as the source, which is erroneous. The plants excluded from this study are listed in Table S2with reasons." The mountainous example is not listed as a reason for any of the 11 excluded plants in table S2. Does that mean that criterion was never used to exclude any plant? If so, then replace with the criteria mentioned in table S2.

Thank you for pointing this out. We removed "mountainous" from text. During the previous revision, we re-examined records to be excluded from our analysis, and noted that elevation should not be used as an evaluation criterion. The sentence now reads "For example, DesalData contains some records indicating that plants in inland regions use seawater as the source, which is erroneous" (lines 116-117). Please note that we needed to exclude another 43 plants because erroneous longitude/latitude information were reported. Altogether 54 plants were excluded in this study.

Lines 103-107: it is not clear yet in the flow of the paper why there is a need to collect data on the distance of desal plants and major cities. Briefly explain

Thank you for this comment. We estimated grid cells that utilize seawater desalination as a primary permanent water resource within the SDM. In many cases, seawater is transferred to desalination plants located near the shore, and freshwater is then further transferred to the cities. We considered that the maximum distance that seawater is transported through desalination is the sum of the distance from the seashore to plants, and from plants to cities. An explanation of this is added in lines 199-200.

Line 149: By inputting ..." rephrase the sentence. replace 'inputting' with 'using', 'extracts' with 'identifies', and 'that' with 'where'. also specify what gridded global maps are used as inputs. Also what conditions? The sentence needs to be rephrased.

Thank you for this comment. This is now rephrased as "SDM is a set of climatological, geographical, and socio-economic conditions to estimate the spatial extent where seawater

desalination is likely used. SDM identifies the grid cells where all of the conditions shown below are met. We call these grid cells the Area Utilizing Seawater Desalination (AUSD)." (lines 224-226)

Line 153: delete 'latest'

Thank you. This has been deleted.

Lines 155-157: replace "but it is not necessarily true that desalination plants are located" with 'but desalination plants are not necessarily located'

Thank you. We rephrased this accordingly.

Lines 162-164: the three conditions stand out as subjective at best. It seems that it would make more sense to discuss the data analysis first to justify the selection of these criteria

The three conditions were derived through extensive data analyses, as shown in the text of the draft article. We initially adopted the standard IMRAD (Introduction, Methods, Results, and Discussion) style, such that the model description (Section 2.4 in Methods) comes before the data analyses (Section 3.1 in Results and Discussion). As you mentioned repeatedly, this is a bit confusing. We accordingly moved Section 3.1 to between Sections 2.3 and 2.4.

Line 167: delete "Conditions B, C, A correspond to each."

Thank you. This has been deleted.

Lines 174-176: again these seem random assumptions without sufficient justification to their logic. Where is the 30-80% coming from? The tendency to promise info to come later in the manuscript makes the manuscript harder to follow.

Assumption C (30–80%) was derived from the data analysis explained in Section 3.1. As stated above, we moved Section 3.1 to before Section 2.4 (model description) to allow for an explanation of the model prior to presenting results, which we hope improves the clarity.

Line 168: "data used and spatial resolution of interest, hence modified if SDM is applied in a different

simulation settings." – not sure what this really means. Rephrase.

We had intended to communicate that thresholds are determined by data analyses, and are dependent on both the dataset used and the spatial resolution. This part now reads: "The thresholds are dependent on the climatological data used and spatial resolution of interest; hence, is expected to change if SDM is applied with different data at a different resolution." (lines 250-252). We hope this improves the clarity.

Line 206: replace 'was installed' with 'were installed'

Thank you. This has been corrected.

Line 245: "We approximated it to 30-80% and set as Assumption C." why not use 28-77% as shown by the data?

We rounded up 28% and 77% to 30% and 80% as the numbers were derived from limited records of data (only 5 records, as shown in Table 3). We therefore assumed that such rounding best reflected the limited nature of the dataset.

Figure S3: Are you accounting for the diminishing purchasing power of money over time (discount rate) when accounting for the 14000$/person threshold? The SSP3 should be a 'poor' world with high population and difficulties to both mitigate and adapt.

All GDP projections used in this study are in units of 2005 USD PPP. The rate of inflation (including discount rate) was excluded from the projections. We agree that SSP3 results partially contradict scenarios deriving from its narrative (low income and difficult to adapt to and mitigate climate change). We do note that country-based GDPPC grows steadily and considerably in all SSP scenarios. Thus, SSP3 is poorer and more fragmented than the other SSPs, but is still richer than the present day. This explanation has been added to the discussion in lines 146-148 and 400-404.

Table S2: what does it mean that 'lat/long data collapsed'? this does not make sense to me. Are you saying these points (lat/long) fall outside of the country boundary (Spain in this case)? Or something else.

Thank you for picking up this point. This sentence has now been phrased as "apparent

inconsistency in location records." For example, longitude and latitude for the Minera Escondida Plant in Chile are actually the location of a hotel in Santiago. We hope this change improves the clarity.

Reviewer 2

I have reviewed the previous version of the manuscript (Reviewer #2). After reading though the revised version of the manuscript, I find the paper much improved and that the authors responded mostly well to my previous concerns. I appreciate that the authors have implemented all of my comments.

We thank you again for your time and valuable suggestions towards improving our study.

As I mentioned in the previous review, this study is novel developing a new desalinated water use scheme, which can be integrated with a large scale water model. This is very beneficial for large-scale hydrological community.

As before, in principal, I would recommend the paper to be considered for publication in HESS. I have a few outstanding comments that the authors may consider to incorporate before submitting a final version.

1. In Abstract, "…(0.002–0.019% of the global total GDP)…", is used, also in in supplement Table S5, all is relative to global total GDP. I don't really find these numbers very useful. I think they could be more relevant if you could specify per country (% of the country total GDP) for both cases (Abstract, Table S5). Costs and GDP are very region-specific so in many cases global cost and global GDP doesn't correspond each other well. Cost is always attributed to a region and associated GDP (not global).

Thank you for this comment. We reported both global and regional numbers in the revised text. Additionally, Table S6 (former Table S5) reports the fraction of desalination cost to GDP for eleven regions and globally. Only global numbers are shown in the Abstract, due to space limitations. We also note that the unit cost of desalination is reported globally in our study (i.e. as stated in lines 427-428).

2. Figure 6. Thank you for your efforts. But I don't see almost any differences for SSP1, 2, and 3.

Can you zoom in a little bit? Or maybe put subpanels?

> We again studied some alternative graphic expressions, but could not come up with a new effective one. The differences among SSP1, SSP2, and SSP3 are less than 200 grid cells out of 67,000 total land grid cells. In any graphic expression, it is difficult to see the difference very clearly and efficiently. Therefore, we described the key differences of SSPs in lines 374-376.

Reviewer 3

The authors of the revised manuscript present a model for areas and volumes of use of desalinated seawater (SDM) that is proposed to be suitable for incorporation into global hydrological models (GHMs) to estimate current seawater use (2005) but in particular to derive scenarios of future use of desalinated water. They also generated three scenarios for the time periods 2011-2040 and 2041-2070 based on the global Shared Socioeconomic Pathways (SSP) scenarios of future population and GDP.

In my opinion, the capacity of the developed SDM for simulating current/historical use of desalinated seawater is so poor such that it is not meaningful or sensible to include the model in GHMs. SDM predicts the existence of desalinated seawater use only as a function of aridity, GPD and distance to ocean. Table 2, a comparison of Fig. 1 and Fig. 4 as well as Fig. 6 show that the assumption that desalinated seawater is used only in very arid grid cell with an aridity index of 0.075 or less precludes the simulation of existing seawater use in Spain and Israel as well as in coastal areas with desalination plants in Libya, Australia, the USA, India and China even in the future (see comp. Fig. 6e and f to 6a-d). While with this aridity criterion, the current bulk of desalinated seawater that is produced on the hyperarid Arabian peninsula is captured, this does not make the model suitable for simulating the extension of seawater desalination to other less arid coastal areas that may be expected (and has been observed) with increasing water stress and decreasing cost of desalination.

> First of all, we are grateful for your time spent reviewing our manuscript, as well as your insightful comments and constructive criticisms. We have further improved our model and interpretation of the results. In the revised paper, we added an auxiliary model that largely improves the reproducibility in desalination plant distribution. We also revised the interpretation of the original and auxiliary models, which is summarized below.
> First, our model simulation is markedly improved. For example, the total desalination water production in nine major countries is 3.0 km$^3$ yr$^{-1}$, where our grid-based simulation using

multi-disciplinary and independent datasets gives 2.4 km$^3$ yr$^{-1}$ (Table 2). Our SDM is both intentionally simple and robust. It could be improved by adding local details, but such specificity could lose applicability across temporal and geographic scales. We also note that detail is lacking from even the most recent global scenarios on which the model is based.

Second, we developed an auxiliary model (SDM2) with a different set of conditions from the original model (lines 237-241 and 272-284). One of the most important conditions of the original model (SDM) is that the Aridity Index must be less than 8% (7.5% in the previous manuscript; the threshold was updated after an intensive data quality check), to account for the fact that major desalination plants are concentrated in hyper-arid climates where municipal and industrial activities are largely dependent on desalinated water. We believe this is a valid and useful predictor, but it belies the fact that a considerable number of desalination plants are beginning to be constructed in modestly arid regions. In the revised paper, we proposed SDM2 with a relaxed Aridity Index threshold and added a new water scarcity threshold (discussed in detail below). SDM2 enabled us to reproduce the geographical distribution of desalination plants much better than the original model, particularly for the Mediterranean coasts. This model now represents existing seawater desalination in Spain and Israel, as well as in coastal areas of Libya, Australia, USA, and China (compare Figure 1 and Figure 4b).

Third, we revised the interpretation of the model and the simulation results (lines 237-241 and 310-323). AUSD estimated using the SDM shows the geographical extent of regions that are chronically dependent on desalination. This is the case for a number of cities on the Arabian Peninsula. Owing to socio-economic changes, this dependence will expand beyond the Arabian Peninsula, as shown in our future projections. We find that the estimated AUSD by SDM2 (shown above) is the geographical extent of regions in which desalination water is used in a supplementary way. By combining the results of SDM and SDM2, we successfully reproduced both the geographical extent and total production of seawater desalination in the world.

Unfortunately, the authors do not describe well how they derived their model and how the results using alternative predictors would be. I think that predictors with a higher predictive capacity should be tested. If it is not possible to derive a more "predictive" model than it is better to refrain from simulating future changes in the use of desalinated seawater. This is the main reason why I think the manuscript should be rejected.

An explanation of how SDM conditions and assumptions were obtained through data analyses is included within the text. Because we adopted an IMRAD (Introduction, Methods,

Results, and Discussion) style in our original submission, the model description (Section 2.4 in Methods) appeared before the data analyses (Section 3.1 in Results and Discussion). This gives the impression that the conditions and assumptions are arbitrary. Thus, we put Section 3.1 before Section 2.4 in the revised text. We hope this makes the format of our experiment clearer.

Second, we developed and tested the SDM2 model that includes an alternative predictor of water resources index (WWR; Raskin et al., 1997) and a water stress threshold of 0.4. Both this index and this threshold are widely accepted, and have been used extensively to assess global water scarcity in the last two decades (e.g. Vörösmarty et al., 2000; Oki and Kanae, 2006). We then relaxed the Aridity Index threshold by as much as 50% (originally 8%); other conditions and assumptions were kept intact. Using the WWR enabled us to incorporate water requirements, or a 'human' aspect into SDM2. This SDM2 expands AUSD to a greater spatial domain including coastal Spain, Israel, northwestern Libya, and islands in Italy, where desalination plants are actually located. Future AUSD spatial extents are shown in Figure S5. Although SDM2 is promising, we could not substitute SDM with this model. Again, the primary objective of this research is to quantify desalination water production. The AUSD estimated by SDM2 includes regions where seawater desalination is used for supplementary water, necessitating information on the availability of other water sources (e.g. river water, groundwater, and others). This demonstrates the need to couple SDM2 and H08 in future research. This model and results are all discussed in lines 237-241, 272-284, and 310-323.

Third, we are aware of other potential predictors, particularly the importance of economic factors. Whether seawater desalination becomes a practical option or not is determined by its production cost relative to that of other water sources. To estimate the cost of water of various sources is challenging because we need to consider seasonal/inter-annual variations in water availability, water quality, regulations, and many other aspects. Including such aspects in global hydrological models is highly challenging, if not infeasible; taking steps to do so in future work will be possible given the progress and improvements in global hydrological models (lines 518-519 and 524-527).

In addition, the three derived scenarios are driven by climate (change), GDP change and domestic and industrial water use change only, with future new grid cells with desalinated seawater use mainly being determined by the different GDP developments in the three scenarios. In the spirit of scenario development, it would have been interesting to represent the different storylines of the SSPs (sustainability vs. middle of the road vs. fragmentation) by e.g. favoring waste water treatment over desalination in the sustainability scenarios SSP1.

Thank you for this comment. Altering water sources and technology by SSP is quite interesting. We added the points you made: we only used GDP information of SSPs in this study to project future expansion in desalination use. Utilizing qualitative (narrative) scenarios would enhance scenario consistency (lines 527-530). We believe this study is a first important step toward such advanced studies. Please note that future projections in industrial and municipal water use fully incorporated both the quantitative and the qualitative SSP scenarios (Hanasaki et al. 2013a,b).

Finally, the presentation of the research even in the revised version is still only fair. For example, neither in the abstract nor in the introduction is it mentioned that they only simulate desalinated sea water use for domestic and industrial uses. The relationship between aridity index and the existence of desalination plants which is central to the model is not shown while a global map of the aridity index is (lines 214-217). It is not discussed what can be learned from then new panels Fig. 6e and 6f. There are many typos, and the manuscript would require language editing.

Thank you for your valuable comments. We have gone through the text and revised it thoroughly to improve the presentation. First, we mentioned the focus on domestic and industrial water in the Abstract (lines 20-21) and Introduction (lines 88-89). Second, we elaborated what we can learn from Figure 2 and Figures 6e-f in lines 184-193 and lines 380-389, respectively. Finally, two professional editors proofread the text. We thus strongly believe that the quality of the paper and presentation of results have been significantly improved through these actions.

References

Hanasaki, N., Fujimori, S., Yamamoto, T., Yoshikawa, S., Masaki, Y., Hijioka, Y., Kainuma, M., Kanamori, Y., Masui, T., Takahashi, K., and Kanae, S.: A global water scarcity assessment under Shared Socio-economic Pathways – Part 1: Water use, Hydrol. Earth Syst. Sci., 17, 2375-2391, 10.5194hess-17-2375-2013, 2013.

Hanasaki, N., Fujimori, S., Yamamoto, T., Yoshikawa, S., Masaki, Y., Hijioka, Y., Kainuma, M., Kanamori, Y., Masui, T., Takahashi, K., and Kanae, S.: A global water scarcity assessment under Shared Socio-economic Pathways – Part 2: Water availability and scarcity, Hydrol. Earth Syst. Sci., 17, 2393-2413, 10.5194/hess-17-2393-2013, 2013.

Oki, T., and Kanae, S.: Global hydrological cycles and world water resources, Science, 313, 1068-1072, 2006.

Raskin, P., Gleick, P., Kirshen, P., Pontius, G., and Strzepek, K.: Comprehensive assessment of the

freshwater resources of the world, Stockholm Environment Institute, Stockholm, Sweden, 1997.

Vörösmarty, C. J., Green, P., Salisbury, J., and Lammers, R. B.: Global water resources: Vulnerability from climate change and population growth, Science, 289, 284-288, 2000.